# Deep learning the *cis*-regulatory code for gene expression in selected model plants

Fritz Forbang Peleke [1,7], Simon Maria Zumkeller [2,3,7], Mehmet Gültas [4], Armin Schmitt [5,6] & Jędrzej Szymański [1,2,3] ✉

Elucidating the relationship between non-coding regulatory element sequences and gene expression is crucial for understanding gene regulation and genetic variation. We explored this link with the training of interpretable deep learning models predicting gene expression profiles from gene flanking regions of the plant species *Arabidopsis thaliana*, *Solanum lycopersicum*, *Sorghum bicolor*, and *Zea mays*. With over 80% accuracy, our models enabled predictive feature selection, highlighting *e.g.* the significant role of UTR regions in determining gene expression levels. The models demonstrated remarkable cross-species performance, effectively identifying both conserved and species-specific regulatory sequence features and their predictive power for gene expression. We illustrated the application of our approach by revealing causal links between genetic variation and gene expression changes across fourteen tomato genomes. Lastly, our models efficiently predicted genotype-specific expression of key functional gene groups, exemplified by underscoring known phenotypic and metabolic differences between *Solanum lycopersicum* and its wild, drought-resistant relative, *Solanum pennellii*.

Regulation of gene expression relies on the complex interaction of proteins and nucleic acids, from DNA to RNA[1]. One of its key mechanisms in gene regulation is the control of transcription by *cis*-regulatory elements (CRE), which are short DNA sequence motifs within the gene's proximal genomic region that are recognised by transcription factors (TF)[2]. On the transcript level, RNA processing determines turnover. This includes intron-splicing[3], stabilisation of RNAs by mRNA-capping[4] and poly(A)-tailing at the RNA 3' end, to name a few[5]. Accordingly, the regulation of gene expression is mediated by a nucleotide sequence code recognised and bound by protein factors. These interactions are interdependent and are referred to as the gene regulatory network (GRN). Current experimental molecular biology techniques can reveal only a portion of its underlying nucleotide code, as they are reductionistic by default. Consequently, the exploration of the plant gene regulatory code can benefit from a holistic approach like deep learning.

There are different experimental methods available to study the interactions of proteins and nucleic acids. These require profound characterisation of, e.g. TFs. Chromatin immunoprecipitation (ChIP) sequencing is the method of choice to determine protein-DNA interactions of TFs[6]. The identification of putative CREs is restricted in ChIP-sequencing, however, because chromatin structure and protein–protein interactions may prevent TFs from binding and, consequently, limit the identification of CREs[7–9]. Other methods, like in vitro microarray binding assays, can determine TF-DNA interactions more specifically, but their application is technically restricted to well-characterised model species[10]. If established for the organism of interest, such methods are well suited to demonstrate the distinct interactions of TFs and CREs. Linking these interaction studies to specific expression patterns of a gene, however, would require further experimentation.

[1]Leibniz Institute of Plant Genetics and Crop Plant Research (IPK), Corrensstraße 3, D-06466 Seeland, OT, Gatersleben, Germany. [2]Institute of Bio- and Geosciences, IBG-4: Bioinformatics, Forschungszentrum Jülich, D-52428, Jülich, Germany. [3]Cluster of Excellence on Plant Sciences (CEPLAS), Heinrich-Heine-Universität Düsseldorf, 40225 Düsseldorf, Germany. [4]Faculty of Agriculture, South Westphalia University of Applied Sciences, Soest 59494, Germany. [5]Breeding Informatics Group, University of Göttingen, Göttingen 37075, Germany. [6]Center of Integrated Breeding Research (CiBreed), Göttingen 37075, Germany. [7]These authors contributed equally: Fritz Forbang Peleke, Simon Maria Zumkeller. ✉e-mail: szymanski@ipk-gatersleben.de

With the increasing amount of genomic data, machine-learning algorithms can be employed to annotate and functionally characterise CREs[11,12]. Integrative approaches capable of systematic investigation of the sequence-to-regulation relationships across multiple plant species and regulation domains are needed[13]. Among these, deep learning (DL) represents the most versatile and increasingly powerful framework that recently enabled significant breakthroughs in e.g. computer vision[14], natural language processing[15] and protein structure prediction[16]. Among DL models, convolutional neural networks (CNN) are particularly effective in the classification, segmentation, and feature extraction of image data and have been already successfully implemented on genetic sequence data. For instance, the Enformer model combined convolutional with multi-head attention layers in a neural network for improved integration of the effects of the distal and proximal regulatory elements in human genomics sequences and efficient gene expression prediction[17]. In *Zea mays*, the expression of genes was successfully modelled as a binary classification task solely relying on a CNN architecture but pointing out the potential overfitting of models to gene family-specific features[18].

Despite these methodological advances, the genome-scale identification and annotation of *cis*-regulatory sequence features across multiple plant species is still an uncharted area. Therefore, in this study, we aim at the systematic identification of gene regulatory sequences and annotation of their function in terms of their effect on gene expression. We investigate the conservation of regulatory sequences and their function across the four model plant species *Arabidopsis thaliana, Solanum lycopersicum, Sorghum bicolor and Zea mays*. We classify gene expression states using a large resource of short-read transcriptome data with convolutional neural networks that scan raw sequences as automated motif extractors. Finally, we demonstrate the application of our models for functional annotation of genetic variation and metabolic pathway activity prediction in domesticated and wild tomato accessions.

## Results

### Data resource

In the first step, we generated a data resource serving as an input for training the model. For that purpose, we used the genome assemblies and annotations of *Arabidopsis thaliana, Solanum lycopersicum, Sorghum bicolor* and *Zea mays* from the Ensembl Plants database (v52). For each gene of each species, 500–3000 nt upstream and 100–700 nt downstream of the transcription start site (TSS) and 100–700 nt upstream and 500–3000 nt downstream of the transcription termination site (TTS) were extracted (Fig. 1a). The selected gene flanking sequence regions were supported by previous studies on plant gene core promoter elements and their conservation and were also sufficient for expression prediction in *Z. mays*[18–20]. The gene flanking sequences were one-hot-encoded to serve as input to the CNN model. For each species, we generated profiles from short-read transcriptome data obtained for leaf and root tissues from publicly available transcriptome experiments under equal conditions (Supplementary Data 1). Estimated gene expression levels per tissue were then classified as low, medium or high based on the lower and upper quartiles of the distribution of the log-transformed transcript

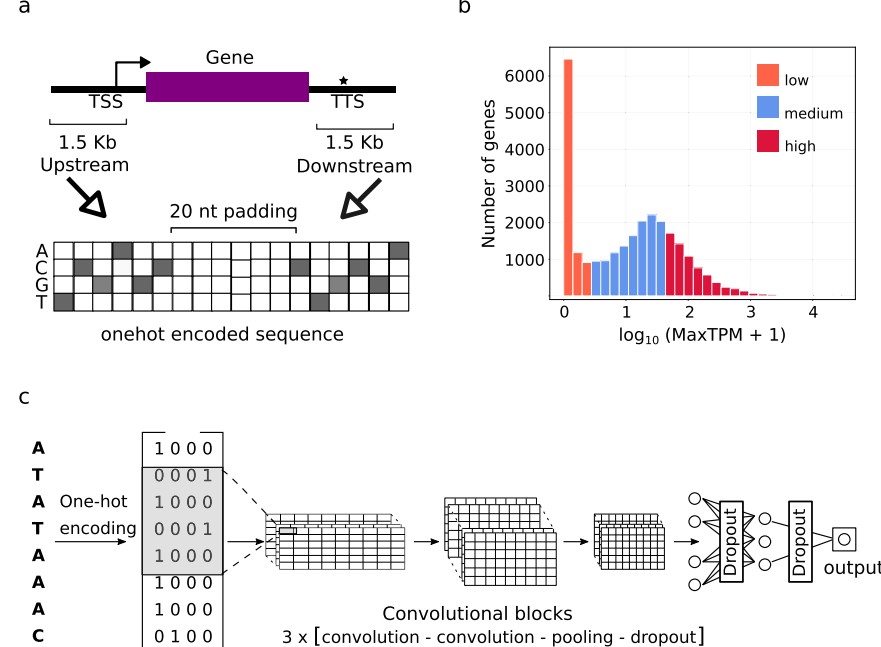

**Fig. 1 | Gene expression prediction models required the extraction of proximal gene sequence from crop plant reference genomes, estimation and classification of transcript levels and nucleotide sequence conversion via one-hot-encoding to generate training data for the modelling in a convolutional neural network.** a Per gene, two proximal regions with a size of 1.5 kbp each were extracted at the transcription start sites (TSS) and transcript termination site (TTS), respectively, fused and separated by a 20 nt padding of Ns. The extracted regions cover 1 kbp of non-transcribed, intergenic region DNA flanking the gene up and downstream, plus 500 bp of each gene transcribed 5′ and 3′ end, covering e.g. UTR regions. DNA regions were extracted 1 kbp upstream and downstream and 0.5 kbp from the annotated gene start and end of genes, respectively. Extracted sequences were converted into matrices by one-hot encoding, separated by a 20 nt padding. b Genes were assigned into low (dark orange), medium (blue), and high (red) expression classes based on the upper and lower quartile of the logMaxTPM distributions (orange, blue and red) exemplarily shown for *A. thaliana*. Histograms for transcript profiles of *S. lycopersicum, S. bicolor* and *Z. mays* are shown in Supplementary Fig. 1. The threshold values for leaf transcript profiles of *A. thaliana, S. lycopersicum, S. bicolor* and *Z. mays* were 0.199, 0.000, 0.153 and 0.113 for the lower and 1.621, 1.051, 1.389 and 1.465 for the higher quartile, respectively (Supplementary Data 1, Source Data). c An end-to-end depiction of model training for one-hot-encoded sequences that were used as training and testing data for the convolutional neural networks (CNN). The CNN architecture consisted of three convolutional blocks, each containing two convolutional layers followed by a pooling and a dropout layer. The final convolutional block was followed by two fully connected layers separated by a dropout layer and a final output layer with sigmoid activation.

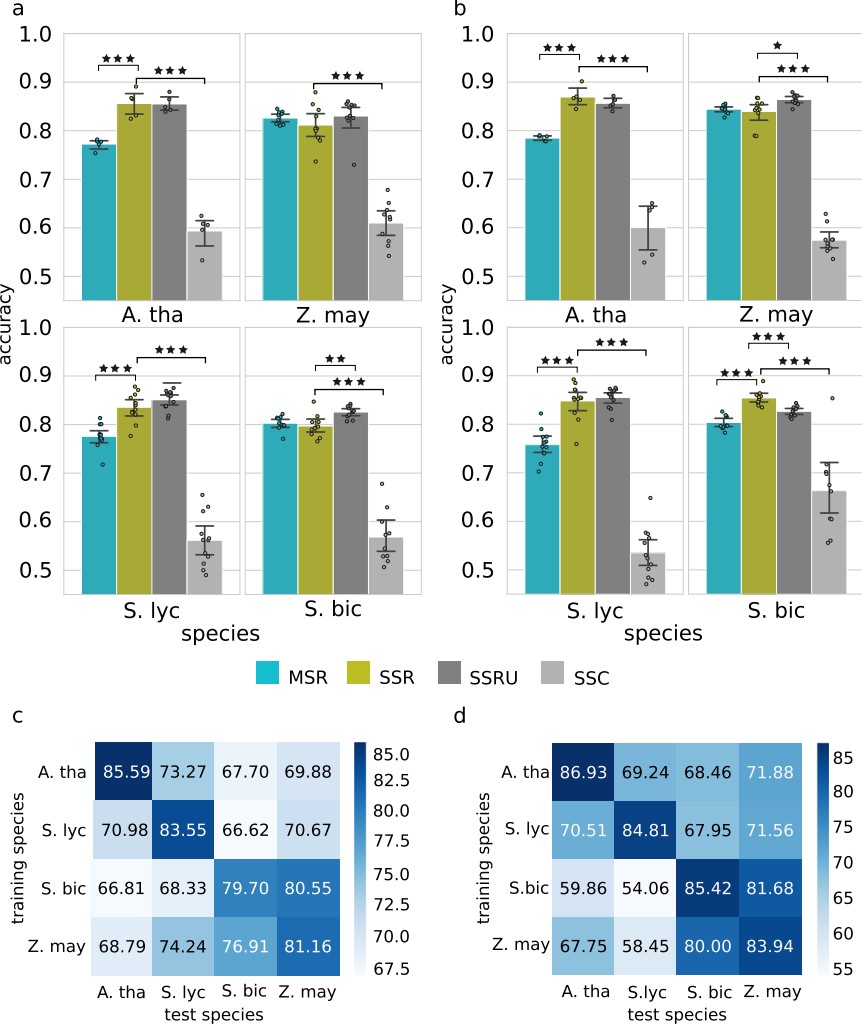

**Fig. 2 | Comparison of predictive performance of deep learning CNN gene expression prediction models for crop plants under varying combinations of training data. a** The leaf model performances were estimated by calculation of prediction accuracy. For each crop plant reference species, *A. thaliana* (A. tha.), *S. lycopersicum* (S. lyc.), *S. bicolor* (S. bic.) and *Z. mays* (Z. may.), four different gene expression prediction models were generated based on varying combinations and variations of the training data. Training consisted of single-species references (SSR), multi-species references (MSR), species-specific references with homologous sequences (SSRU) and shuffled-sequence controls (SSC) (Supplementary Data 5). The error bars represent the 95% confidence intervals; the significance of the two-sided *t*-test is depicted as asterisk (*p* value: * ≤0.05, ** ≤0.01, *** ≤0.001); the number of observations $n = 5$ (*A. thaliana*), $n = 12$ (*S. lycopersicum*), $n = 10$ (*S. bicolor*) and $n = 10$ (*Z. mays*) (Source Data). **b** The root model performances were estimated by calculation of prediction accuracy (see caption for sub-figure **a**). **c** Cross-species performances for leaf models were estimated by predicting expression profiles of other species using species-specific models. The highest cross-species accuracy was 80.55%, testing the SSR model of *S. bicolor* on *Z. mays*. The lowest estimated accuracy was 66.62% testing the SSR model of *S. lycopersicum* on *S. bicolor*. **d** The highest cross-species accuracy for the models generated with root transcript profiles was 81.68%, testing the SSR model of *S. bicolor* on *Z. mays*. The lowest estimated accuracy was 54.06%, testing the SSR model of *S. bicolor* on *S. lycopersicum*.

per million values (logMaxTPM) (less than 25% percentile, between 25% and 75% percentile, above 75% percentile, respectively) (Fig. 1b and Supplementary Fig. 1).

## Model architecture and training strategy

Our approach builds on the CNN model architecture proposed previously for the pseudogene model with small variation[18] (Fig. 1c). The three convolutional blocks used, each composed of two convolutional layers, have proven to efficiently capture sequence features of different scales and complexity within the flanking sequences. As in CNNs used for image classification, the use of multiple layers in each block enables parallel capturing of multiple features of the input DNA sequence by the first convolutional block, and of the output of the previous convolutional block for the blocks two and three. Finally, the output of the convolutional layers is being integrated by the fully connected layer block for expression prediction. In contrast, to the 2D

convolutional and max pooling layers used by Washburn and colleagues[18], we used 1D layers as it has been the choice in more recent studies inferring gene expression or protein interaction from DNA sequence using deep learning, e.g. the Enformer or BPNet architecture[17,21].

To train the CNN model, we focused our analysis on genes below and above the lower and upper quartiles of the logMaxTPM distribution, respectively. Accordingly, the CNNs were trained as binary classifiers to predict genes as either lowly (below the lower quartile) or highly (above the upper quartile) expressed. Dividing the data into more classes as well as training a regression model resulted in considerably lower accuracies (Fig. 2a). CNNs were trained using chromosomal level cross-validation, in which for each iteration, genes located on one of the chromosomes were used as a validation set and the rest for training. As outlined by Washburn and colleagues (2019)[18], to prevent overestimation of model performance due to overfitting

evolutionary relatedness, genes in the validation set with homologues in the training set were omitted in the validation set. This assures that the training is not biassed by the sequence homology within gene families.

The performance of the models trained on different input sequence sizes remained similar between 500 and 3000 nt for the promoter and terminator sequences (Supplementary Fig. 2). We observed a species-specific drop in performance only for the UTR sequences below 300 nt. Thus, taking also into account the distribution of the length of annotated UTR regions across the four species of interest (Supplementary Fig. 3) and the reduced interpretability of models working on large inputs, we decided on the uniform range for the up- and downstream regions for all four species. Accordingly, all genes were represented with a 1000 nt promoter and 500 5′UTR covering the upstream region and 500 nt 3′UTR and 1000 nt terminator for the downstream region as a choice. The combined 3000 nt sequences provided high performance and a considerable buffer to capture possible differences in distributions of the core regulatory elements in genomes of significantly different sizes. The detailed analysis of the predictive power of general gene model features, such as UTR length or GC content, in gene expression prediction, is provided in Supplementary Materials (Supplementary Fig. 4 and Supplementary Note 1).

## Predicting plant gene expression from the sequence of gene flanking regions− model performance

In the first step, we trained the CNN models per species with $n = \#$chromosomes cross-validation rounds. The use of chromosomes for splitting the validation set ensured that the set was not overlapping with any of the training sequences and was similar in terms of chromatin accessibility. These single-species reference (SSR) models achieved average accuracies for *Arabidopsis thaliana* $acc_{leaf} = 85.59\%$, $acc_{root} = 86.93\%$ ($auROC_{leaf} = 0.92$, $auROC_{root} = 0.94$), for *Solanum lycopersicum* $acc_{leaf} = 83.55\%$, $acc_{root} = 84.81\%$ ($auROC_{leaf} = 0.89$, $auROC_{root} = 0.90$), for *Sorghum bicolor* $acc_{leaf} = 79.70\%$, $acc_{root} = 85.42\%$ ($auROC_{leaf} = 0.85$, $auROC_{root} = 0.88$) and for *Zea mays* $acc_{leaf} = 81.16\%$, $acc_{root} = 83.94\%$ ($auROC_{leaf} = 0.87$, $auROC_{root} = 0.90$) (Fig. 2a, b, Supplementary Fig. 5, and performance of all models and controls with F1 scores provided in Supplementary Data 2).

This result showed the extent of flanking sequence-determined gene expression in leaves and roots that could be learned by a simple CNN and, thus, likely represents rather an underestimation of the real value. The varying size of the training sets between SSR models did not directly reflect the model performance, indicating that it is not a major limiting factor. RNA-seq profiles for the same species and condition can differ due to technical and experimental variations. Accordingly, we exemplarily trained and cross-evaluated leaf SSR models from multiple comparable RNA-seq experiments of *A. thaliana*. These achieved high test performances supporting our methods' reproducibility (Supplementary Fig. 6). Next, we trained multi-species reference (MSR) models using a leave-one-out procedure; each time training the model on three of the species and evaluating it on the fourth. The MSRs showed a similar performance as the species-specific SSRs, showing that the predictive sequence features are conserved (Fig. 2a, b). The shuffled-sequence controls (SSC) retaining the nucleotide composition exhibited close to random classification performance for all species, highlighting the existence and the role of localised sequence features as gene expression predictors. The observed model performance for the tissue-specific genes is significantly lower than for a comparable random set of labelled genes, indicating the tissue-specific regulation is orthogonal to the regulatory signature identified by the model that is trained and generalisable on all genes (Supplementary Fig. 7).

This stayed in concordance with variation of prediction accuracy between functional categories (Supplementary Data 3). Biological processes showing low or no expression in the analysed tissues of interest exhibited lower accuracy (e.g. bin-13.3 'Cell division.meiotic recombination'; average acc = 0.68) than those highly expressed (e.g. bin-2.3 'Cellular respiration.tricarboxylic acid cycle'; average acc = 0.87). The correlation between the average prediction accuracy and the average expression of functional categories reached Spearman's $\rho = 0.76$ and $p$ value <2.2e-16. Finally, we tested the SSR models in terms of their prediction accuracy in species they were not trained on, in what we called a cross-species prediction (Fig. 2c, d). The best cross-species prediction accuracy was observed for the *Sorghum bicolor* SSR_leaf and SSR_root model, with 80.55% and 81.68%, respectively, using *Zea mays* sequences for prediction. The worst performance was observed for the *Solanum lycopersicum* SSR_leaf and *Sorghum bicolor* SSR_root model predictions on *Sorghum bicolor* and *Solanum lycopersicum* sequences with 66.62% and 54.06%, respectively (Supplementary Data 2). The cross-species prediction accuracies indicate that the model's performance may depend on the species' evolutionary relationship, with higher cross-performance for more closely related species.

The significant drop in cross-species prediction performance in comparison to the SSR and MSR models indicated that the gene expression in each species is determined by two classes of sequence features: (a) species-specific and (b) conserved across multiple species. While the SSR models learn the combination of them and are hardly generalisable across species, the MSR models seem to capture the conserved features and generalise well.

## Identification and characterisation of predictive sequence features

Identification of the location and specific sequence features associated with gene expression requires interpretation of the trained models. Here, we used the Deep Learning Important Features (DeepLIFT) algorithm, which enables the interpretation of convolutional neural networks trained on genetic sequences by providing nucleotide-resolution importance scores across input sequences[22]. The scores are calculated from the models by backpropagating the importance of each neuron to each nucleotide of the input sequence. Subsequently, the importance of each nucleotide is represented as a score that can be positive, negative, or zero, corresponding to the prediction of the training classes (high expression and low expression or irrelevant for prediction), respectively. Averaging obtained scores across the input sequences revealed conserved patterns, called salient regions or saliency maps. We observed that for all species, the most salient regions of the flanking sequences were those proximal to the TSS and TTS containing the 5′UTR and 3′UTR, respectively (Fig. 3a). The salient regions received predominantly positive scores indicating the presence of expression-associated features, rather than those with deleterious effects. While the general distribution of DeepLIFT importance scores was conserved across the species, some differences also occurred. *Solanum lycopersicum* exhibited a much higher saliency region upstream of the TSS than the other species. *Zea mays* and *Sorghum bicolor*, on the other hand, exhibited low saliency of the sequences downstream of the TTS.

The DeepLIFT importance scores were further used to extract expression-predictive sequence motifs using the Transcription Factor Motif Discovery from the importance scores algorithm (TF-MoDISco)[23]. Here, partitioned sequences, called seqlets, with similar sequence and concordant importance scores were aggregated into contribution weight matrices (CWMs). The CWMs encoding is analogous to the position weight matrix. However, despite representing a positional nucleotide frequency, CWMs contain the importance scores of nucleotides associated with the prediction expression classes. Accordingly, we will refer to CWMs identified with the TF-MoDISco as expression-predictive motifs (EPM) in the following sections. The occurrence of each EPM showed a general specificity

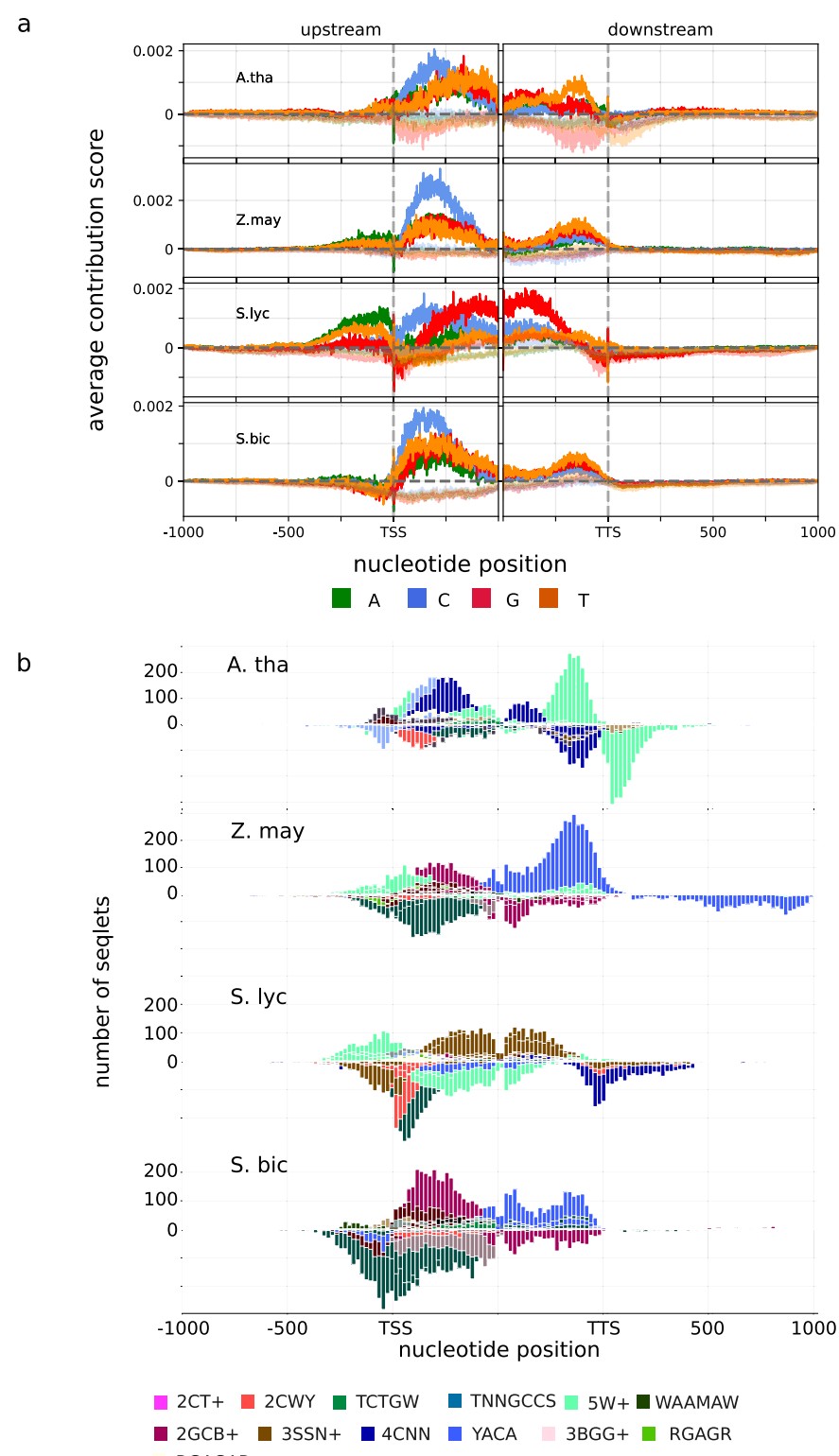

**Fig. 3 | Interpretation of the gene expression-predictive models shows important predictive nucleotides are located mainly in the regions near the TSS and TTS across different crop plants for the leaf SSR models. a** Average per-base importance scores computed with DeepLIFT[22] for *A. thaliana, S. lycopersicum, S. bicolor* and *Z. mays* SSR leaf models. Importance scores were averaged across validation sequences depicting the average importance of regions with the 3000 nt long sequences. **b** Per-base importance scores were used for motif generation with TF-MoDISco[23]. The resulting expression-predictive motifs (EPMs) share consensus sequences following the extended IUPAC nucleotide code (Supplementary Data 4) and position-specific occurrences around the TSS and TTS region.

towards the 5'UTR and 3'UTR, with individual EPMs showing EPMs-to-region specificity that we, from here on, refer to as preferred ranges. Similarly, to the saliency map (Fig. 3a), the positional distribution of EPMs exhibited both conserved and species-specific patterns

(Fig. 3b). Varying the sizes of UTR and Promoter/Terminator regions did not change the resulting saliency map for the chosen genomic ranges between 500 nt and 3000 nt (Supplementary Fig. 8). However, shortening the UTR input ranges below 500 nt had an impact

both on the saliency map and model performance for SSR leaf models of all four reference species.

## Expression predictive motifs exhibit sequence and positional conservation

An EPM has a sum positive or negative importance score associated with high or low predicted expression classes (Supplementary Data 4). Accordingly, the EPMs were divided into two metaclusters by TF-MoDISco depending on their positive or negative importance scores calculated by DeepLIFT. Consequently, the two metaclusters are associated with the prediction of either high or low rates of gene expression and were labelled as 'p0' (149 EPMs) and 'p1' (111 EPMs) for leaf data, respectively. Congruently, the sum importance score of one gene relates to the predicted class. Across all four species and models the sum of importance scores per gene flanking regions were positive when it was predicted to be highly expressed and vice versa. For example, two CT hexamer repeats can be found downstream of the TSS of *Arabidopsis thaliana* gene *AT1G01650*, correctly predicted for high gene expression with positive importance scores of 3.856 (Fig. 4a). These regions match the SSR leaf model EPM epmArth-S019-p0m06 with a sum importance score of 0.31 (Fig. 4b). This EPM displays 99% similarity ($p$ value = 0.0002) to the transcription factor binding site (TFBS) of *A. thaliana* BPC5 found by comparison to the JASPAR2022 plant database (MA1403.1)[24] (Fig. 4b, Supplementary Data 4). BPC5 has been characterised to bind the GA/CT-repeat binding motifs and belongs to the BASIC PENTACYSTEINE (BPC5/BBR/BPC) transcription factor family[25].

In total, we have identified 260 EPMs (520 with reverse complement versions). This includes 39 EPMs in *A. thaliana*, 26 EPMs in *S. bicolor*, 30 EPMs in *S. lycopersicum* and 36 EPMs in *Z. mays* for the SSR leaf models and 30, 35, 37 and 27 for the MSR leaf models, respectively (Supplementary Data 4). Hierarchical clustering of the 520 leaf EPMs using the Smith−Waterman algorithm identified 17 multi-species clusters of similar motifs from the SSR and MSR leaf models (Fig. 4c and Supplementary Fig. 9). These 17 clusters included 494 of 520 EPMs, while 26 EPMs remain without association by similarity. The smallest and largest clusters include 8 and 80 different EPMs, respectively. In line with the analyses of testing model performances across different tissues, the comparative analysis of EPMs from leaf and root SSR models exemplarily tested for *A. thaliana* and *Z. mays* show that EPMs from both tissues are highly similar and fall into the same clusters (Supplementary Fig. 10 a, b).

In the following sections we name these EPM clusters after their consensus sequence alignment according to the IUPAC nucleotide code with the minimum number of repetitions as a prefix (Fig. 4b and Supplementary Data 5). For example, cluster 2CT+ (CTCTCT or AGA-GAG in reverse complementary orientation) features EPMs with at least two or more (+) cytosine-thymine dimers. Of the now 18 different clusters, we identified 15 clusters that have at least one EPM with significant similarities to a TFBS in the JASPAR2020 DB ($p$ value <0.001 and e-value <0.01)[24] (Fig. 4c and Supplementary Data 4). Interestingly, we identified 26 different TFBS in twelve clusters (2CT+, CGNCGT, WAAMAW, 2GCB+, GSRGV, 2CWY, TNNGCCS, 5W+, YACA, TCTGW, CTAG and 3SSN) that have a very high similarity of 95% measured with Pearson correlation coefficient (PCC). Of these TFBSs, eleven are solely matching to EPMs from the SSR leaf models, compared to seven from the MSR, and nine shared ones. This does indicate that the leaf SSR models perform slightly better on the identification of putative TFBSs. The remaining EPM not matching characterised TF binding sites showed similarity to structural gene elements, e.g. within the 5W+ cluster. There are, for example, the U and A-rich regions involved in the selection of mRNA polyadenylation sites[26] (e.g. polyadenylation site "5'-AAUAAA" and epmArth-S030-p0m14) and hexanucleotide U repetitions acting as terminators for the nuclear RNA Polymerase III[27].

## The EMP cluster 2CWY+ is most similar to the TF binding site of *A. thaliana* AGL42

All four leaf MSR models identified EPMs that belong to the 2CWY+ cluster. This cluster uniformity contributed to the prediction of low gene expression with sum importance scores ranging from −0.2 to −0.3 in epmZema-S068-p1m015 and epmArth-*SO57-p1m14*, respectively (Supplementary Data 4). In all four analysed species, 2CWY+ EPMs exhibit a strict positional preference for the TSS and TSS-downstream), and, in some cases, the transcription termination site TTS and upstream-TTS (Fig. 4d). The high conservation of the sequence, the positional preferred range and uniform prediction towards low rates of transcription suggest an evolutionary conservation and regulatory function of 2CWY+ type EPMs. In total, all fifteen EPMs of the identified 2CYW+ cluster exhibited conserved position-dependent predictive performance on gene expression (Supplementary Fig. 9 and Supplementary Data 6). Congruently to their well conservation across species and genomic position, we found EPMs of the 2CWY+ cluster with high similarity to counterparts in exemplarily investigated *A. thaliana* and *Z. mays* root SSR model EPMs, as well (Supplementary Fig. 10 a, b). These findings underline that EPMs represent generalised genomic features across species and tissues. The EPMs of the 2CWY+ clusters, e.g. epmArth-S063-p1m08 or epm-Soly-*SO35-p1m02* (CATCAT), significantly matched the TF binding site (CAYCAT) of AGL42 (also FYF, FOREVER YOUNG FLOWER). AGL42/FYF is a MADS-box transcription factor of the eukaryotic evolutionary old MIKC class closely related to the SUPPRESSOR OF OVEREXPRESSION OC1 (SOC1) which is a gene central to the development of the flower, particularly mediated by gibberellin signalling[28,29]. AGL42 (FYF) interacts with SOC1 and is expressed in the shoot apical meristem during the vegetative phase and during the floral transition where it promotes flowering at the shoot apical and axillary meristems[29]. Here, AGL42 (FYF) actively controls floral senescence/abscission by repressing ethylene responses together with a regulatory network of *FYF*-like genes[30–32].

## The EPM clusters 2GCB+ and 2CT+ are likely TF binding sites but predict low and high gene expression dependent on their genomic position

Like cluster 2CWY+, clusters such as 2GCB+, 2CT+ and YACA feature position-dependent prediction, but contained similar EPMs from different metaclusters (Fig. 4d). For example, eight of fourteen 2GCB+ type EPMs found in *A. thaliana*, *S. bicolor*, *S. lycopersicum* and *Z. mays* that are located within the upstream transcript boundaries and consistently associated to high transcript levels with sum importance scores ranging from 0.31 to 0.46 (Supplementary Fig. 9 and Supplementary Data 4). Conversely, six 2GCB+ type EPMs of *S. bicolor* and *Z. mays* that are prevalent upstream of the TSS and in the 3'UTR region predict low rates of gene expression with sum importance scores ranging from −0.19 to −0.26.

The EPMs of 2GCB+ show significant sequence similarity to the GCC box TF binding sites of the ERF/DREB class of ABA-responsive WRKY-TFs in *Z. mays*[33]. In *S. lycopersicum* the ERF/DREB TFs binding to GCC boxes is a key element of fruit development and ripening signalling[34].

Another EPM cluster with a position-dependent prediction are the 2CT+ types, including epmArth-S003-p0m01, epmArth-S013-p0m06, epmArth-S047-p1m09, and epmSoly-*SO19-p0m09*, identified by root and leaf SSR model of *A. thaliana and S. lycopersicum*, exclusively (Fig. 4 and Supplementary Fig. 10 c). The 2CT+ motifs identified in our study are either localised upstream of the TSS and predict low gene expression levels or are localised downstream of the TSS and predict high gene expression levels. This is further supporting that the EPMs sequence and position is crucial for its use as predictive features across species and tissues.

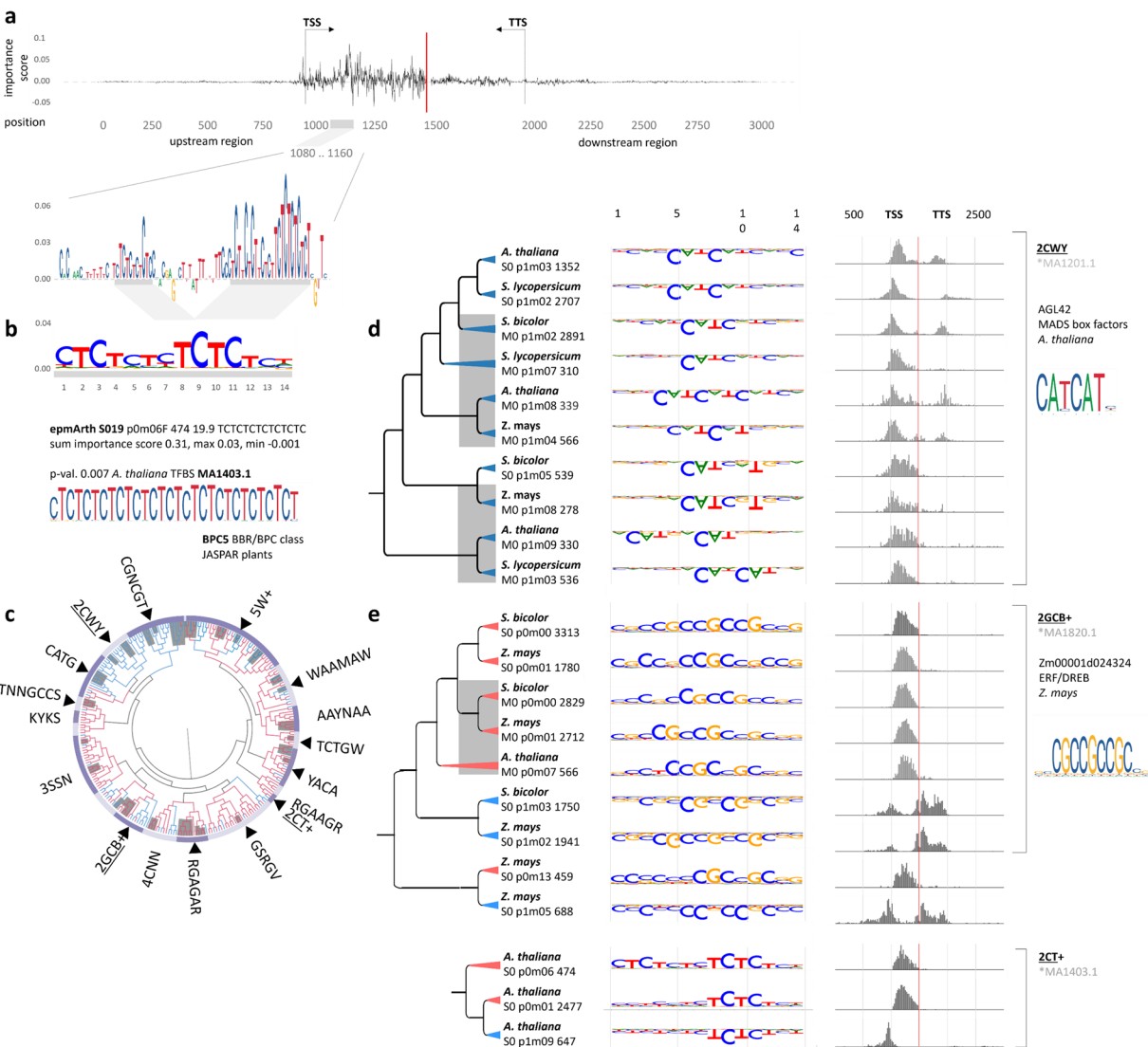

**Fig. 4 | Expression predictive motifs (EPMs) identified by DeepLIFT and TF-MoDISco determined with convolutional neural networks (CNN) trained on single-species reference (SSR) and multi species reference (MSR) models of _A. thaliana_, _S. lycopersicum_, _S. bicolor_ and _Z. mays_. a** Importance scores (IS) in the 1.5 kbp upstream and downstream selected region of exemplarily chosen gene _AT1G01650_ of _A. thaliana_ that is predicted by SSR leaf model with a positive sum IS of 3.86. The region with the maximum IS of _AT1G01650_ lies in the upstream region 1097-1155 bp including two cytosine-thymidine-hexamers (6CT). **b** The 6CT motifs matched with an expression-predictive motif (EPM) that was inferred with TF-MoDISco. For clarity, we propose an EPM nomenclature system that assigns abbreviations to plant species based on their genus and epithet, followed by the model used to produce the EPMs (SSR or MSR). The physiological conditions of the plant are indicated by a number (0 for standard conditions). The predictability of each motif is indicated by a 'p' followed by several 1 s and 0 s for low and high rates of gene expression, respectively. The delimiter is followed by the motif number within the metacluster and its orientation (forward or reverse). Finally, the number of seqlets included, the information content, and the consensus sequences are added at the end of the EPM. For example, epmArth-S019-p0m06 has a sum importance score of 0.31, a maximum importance score of 0.03 and a minimum score of −0.001 (Supplementary Data 4). It has been found three times in the upstream region of _AT1G01650_. In addition, epmArth-S019-p0m06 matched with 99% similarity measured with Pearson correlation coefficient (PCC) and e-value = 0.002 transcription factor binding site (TFBS) of _A. thaliana_ BPC5 of the BRR/BPC class (Supplementary Data 4, JASPAR accession MA1403.1). According to the nomenclature, proposed henceforth, the EPM was identified in _A. thaliana_ (Arth), by the SSR model (S), under standard conditions in leaf (0), predicting high gene expression rates (p0), inferred from 443 seqlets with an information score, indicating nucleotide frequency, specificity, and motif heterogeneity of 19.4, along with its consensus sequence (CTCTCT). **c** The EPMs are assigned into 17 clusters based on similarity using the Smith–Waterman algorithm and manual inspection of the consensus sequences following the alignment. The clusters are named after conserved DNA motifs and indicated by the IUPAC nucleotide code, along with the least number of repeats of motifs (numerals) and potential additions (+). Clusters with EPMs that significantly match TFBSs from the JASPAR database with e-value <0.05 compared using PCC are marked with black triangles. The EPMs identified by the MSR model are highlighted by grey boxes in the dendrogram, while prediction for high and low gene expression is displayed by red and blue branches, respectively. Underlined clusters, with selected representative EPMs, are shown exemplarily in panels (**c**) and (**d**). The complete full-scale version of the dendrogram and the consensus sequence alignment EPMs can be found in supplementary Fig. 9 and supplementary data 5. **d** EPMs of the 2CWY+ cluster uniformly predict low rates of gene expression (blue tips). EPMs of this cluster are identified by the SSR and MSR (grey boxes) models. In contrast to the SSR model, the MSR models identified 2CWY+ motifs for all four reference species. The inverted web-logos show the EPMs negative importance scores, ranging from 0 to 0.05 or −0.05 associated with DeepLIFT and TF-MoDISco metacluster 0 or 1 (p0, p1), respectively. Histograms display the positional preference inferred from the number of seqlets relative to the transcription start and transcription termination sites (TSS, TTS) of each EPM. The EPMs of the 2CWY+ type display significant similarity to the transcription factors binding site of AGL42 (JASPAR accession MA1201.1) of _A. thaliana_. **e** The 2GCB+ and 2CT+ clusters contain EPMs predicting both low and high gene expression rates, identified by the SSR and MSR models, corresponding to the positional occurrence related to the TSS or TTS. Both clusters highly resemble previously determined transcription factor binding motifs for 2GCB+ and 2CT+, with MA1820.1 and MA1403.1, respectively.

## The EPMs of metacluster 0 of *A. thaliana* SSR and MSR leaf model are predictors for high levels of gene expression

We tested the predictive performance of the occurrence of EPMs by mapping those of *A. thaliana* extracted from the leaf MSR and SSR models to *A. thaliana* genes within their respective preferred ranges. We calculated EPMs individual enrichment within their associated expression class and the predicted class for *A. thaliana* leaf SSR and MSR models following a modified formula to determine feature enrichment described by Smet and colleagues[35] (Supplementary Data 6). The EPMs of the *A. thaliana* SSR model associated with the prediction of high gene expression levels range from 1.186 to 2.931 and 1.529 to 3.803 for enrichment among their expression class, respectively. At the minimum and maximum end lies epmArth-S007-p0m03 (4CNN cluster) and epmArth-S013-p0m06 (2CT+ cluster) that can be found in 76 and 94% of cases among genes with high gene expression, respectively. These observations also account for the MSR models, while these features have generally lower scores for enrichment, ranging from, e.g. 0.93 to 2.714 and 0.807 to 2.662 for the expression and predicted class, respectively. Notably, the *A. thaliana* leaf MSR models EPMs associated with high rates of gene expression occur in 68 to 88% of cases among their respective expression class, with epmArth-M0-p0m03 (5 W+ cluster) at the lower and epmArth-M0-p0m11 (RGAAGR cluster) at the higher end, as well. For the clusters with similarities to TF binding sites, as in the case of 2GCB+ and 2CT+, the maximum enrichment scores lie at 2.262 and 3.76. Accordingly, the occurrence of these EPMs of *A. thaliana* leaf SSR and MSR model associated with high rates of gene expression display useful predictors (>80% TPR) for high rates of gene expression and its prediction.

EPMs associated with low expression levels were not enriched among their respective classes, with mostly negative enrichment scores, ranging from −0.573 to −1.202 and 0.409 to −1.160 for the expressed and predicted class, respectively (Supplementary Data 6). All EPMs of metacluster 1 appeared in less than half of the cases within the associated expression class. This shows that EPMs associated with low gene expression by the models do not exclusively occur among genes exhibiting low expression levels and can also be found among ranges of genes of the high gene expression class. Accordingly, the predictive performance of *A. thaliana* EPMs associated with low gene expression might rather relate to specific contextual factors, for instance, the combination or yet unknown features.

## CNN models identify perturbations in gene expression associated with structural variance among sub-species and varieties of tomato

To check if our model predicts differences in expression of gene variants, we used a large collection of genomic structural variants (SVs) for wild and cultivated *Solanum spp*[36]. Due to the application of long-read nanopore sequencing the dataset contained variation data ranging from single nucleotide polymorphisms to large transposon insertions across 15 *Solanum spp*. genotypes.

We used the leaf MSR models trained with *Solanum lycopersicum* as the validation set to generate predictions for a subset of genotypes of *Solanum pimpinellifolium* (PAS014479, BGV006775), *S. lycopersicum* var. cerasiforme (BGV006865, BGV007931 and BGV007989), *S. lycopersicum* processing (M82), *S. lycopersicum* fresh (EA00371, Fla.8924, Floradade and LYC1410), *S. lycopersicum* vintage (PI69588, Brandy-wine, EA00990 and PI303721) and *S. lycopersicum* (ITAG3.0) for reference (Fig. 5a). Neighbour joining-based clustering of the input regulatory sequences for the 15 genotypes showed difference in topology with the hierarchical tree of the predicted expression profiles and remained largely in agreement with the SV-based phylogeny from Alonge and colleagues (Fig. 5a). The gene expression is predicted to be more similar amongst *S. pimpinellifolium* genotypes and the BGV007931 genotype of *S. lycopersicum var. cerasiforme*. Also, clades

of *S. lycopersicum* vintage and fresh form one joint clade, indicating that prediction of gene expression does not strictly follow insertion of SVs across *Solanum* spp. We further investigated orthologous genes that were likely to vary in expression across the genotypes according to predictions of the MSR models and where differential gene expression was shown by Alonge and colleagues (Supplementary Data 7). To characterise the change in gene expression, we mapped motifs from the leaf MSR model of *S. lycopersicum* to the different genotype genes using BLAMM[37] (Supplementary Data 8).

We identified an intersection of 314 genes featuring SVs and log-fold changes in gene expression, determined by Alonge and colleagues, and genes with predicted different expression levels across the fifteen genotypes, according to our model (Fisher exact test value <0.0001) (Fig. 5b). In comparison between genes with predicted homogeneous and differential rates of expression, EPMs appear less often conserved among the differential expressed genes (Fig. 5c). This suggests that the occurrence and conservation of EPMs across different genotypes can be used to further interpret differences in the prediction of gene expression levels. For in-depth analyses, we selected six random examples of genes that belong to the intersection of the 314 genes that all feature mutated (non-conserved) EPMs (Supplementary Data 9). Here, the absence or presence of EPMs coincides with changes in the predicted expression classes.

We chose the genes *Solyc02g08170.4* encoding a putative phosphatidylinositol glycan, class N protein and *Solyc02g080300.3* encoding a glycosyl hydrolase *β*-glucosidase 45-related proteins. Both genes feature SVs in their TSS and TTS regions accompanied by differential gene expression. Further, probabilities of the predicted expression patterns of those genes were independent of their phylogenetic relationship between the cultivars (Fig. 5d, g). There were five and four EPMs, respectively, significantly matching (e-value <0.0001) the TSS regions of the *Solyc02g087170.4* and the *Solyc02g080300.3* (Fig. 5e, h). However, only two and one EPM, for each gene, respectively, were identified in their preferred ranges inferred from the CNN.

According to the probability of gene expression, the gene *Solyc02g08170.4* is predicted to exhibit high (on average of 0.6) and low (on average of 0.35) gene expression in two distinct groups of *Solanum* genotypes (Fig. 5d and Supplementary Data 7). On the sequence level, this difference coincides with a shared 178 bp indel mutation 38 bp downstream of the TSS amongst five of the fifteen *Solanum* genotypes, all predicted for high rates of gene expression (PAS0144795, BGV007989, BGV006865, EA00990 and EA00371) (Fig. 5e, f). Interestingly, disrupting epmSoly-M006-p0m02 via the SV entails the creation of epmSoly-M032-p0m15 now positioned within its preferred range 226 bp downstream of the TSS within *Solyc02g08170.4* 5′UTR (Fig. 5f). This indicates that the CNN predictions depend on the identification of EPMs at their preferred range and that these are weighted differently.

In the case of *Solyc02g080300.3*, an SV indel mutation occurred in the first intron and the 3′UTR of the genotypes PAS014479, BGV06775 and BGV006865 sharing high probability scores compared to the other species (Fig. 5g). Twenty nucleotides upstream of the SV within the first intron two point mutations were found in PAS014479, BGV06775 and BGV006865, as well, hereby matching epmSoly-M009-p0m04 (Fig. 5h, i). This EPM predicts high rates of gene expression and is located downstream of the TSS. Accordingly, its presence explains the prediction of high gene expression for *Solyc02g080300.3* along three of the fifteen genotypes. The probability for higher rates of gene expression is like the one in *Solyc02g087170.4*, where epmSoly-M009-p0m04 could also be identified in its reverse complement orientation.

The prediction of the CNN can be interpreted using DeepLIFT, TF-MoDISco and BLAMM. As a result, we retrieved EPMs that can

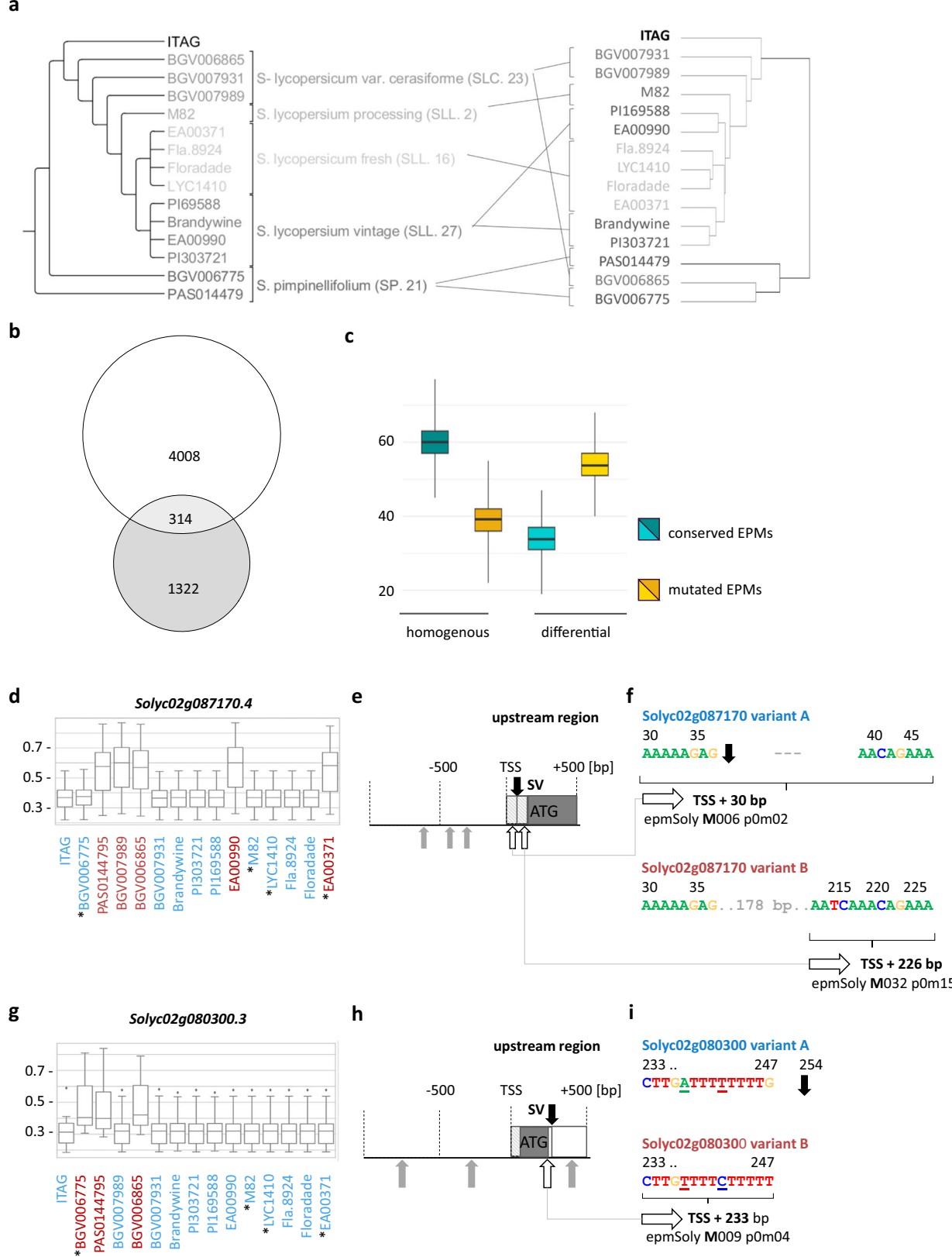

potentially be used for the prediction of rates of gene expression and characterisation of promoter or terminator regions. The CNN, as interpreted using the EPMs, is sensitive to point and indel mutations. In addition, the EPM predictive power is likely weighted differently, as the probabilities differ depending on the identified EPM that is congruent with different importance scores and their predictive performance.

## MSR model highlights known phenotypic and metabolic differences between *S. lycopersicum* and *S. pennellii*

In the next step we evaluated if the MSR model highlights interpretable differences between plant species that were not used for training. For this purpose, we selected the cultivated tomato *S. lycopersicum* and its wild relative *S. pennellii* as examples of highly related species with

**Fig. 5 | Comparison of the predictability of expression-predictive motifs (EPMs) in 15 different *Solanum* genotypes with structural variations (SVs) in the transcription start site (TSS) and transcription termination site (TTS) identified and characterised by Alonge and colleagues (2020)[36]. a** The taxonomic grouping of fifteen *Solanum* genotypes based on SVs inferred by Alonge and colleagues (left) in comparison to hierarchically clustered predictions of gene expression (right) display differences in topology for the groups of S. lycopersicum var. cerasiforme (SLC. 23) and vintage (SLL,27.). **b** There is an intersection of 314 genes between genes with SVs in their upstream or downstream 5 kbp regions, which were detected for log-fold change in gene expression levels across any of the fifteen *Solanum* genotypes, and genes with variances exceeding 0.005 in their predicted probabilities from the *Solanum* MSR leaf model, indicating differential expression across the 15 genotypes. Six random examples were selected from the intersection for detailed examination, as shown in panels (**d**–**i**), with further material provided in supplementary data 9. **c** Examined EPM variation in 15 *Solanum* genotypes, analysing genes with conserved or mutated EPMs alongside shifts in gene expression levels. Genes with homogenous gene expression ($n = 27,993$) showed higher rates of conserved EPMs (blue boxplot), while those with differential gene expression ($n = 2053$) exhibited higher rates of mutated EPMs (yellow box plots). Gene expression heterogeneity was determined based on MSR leaf model probability, using variance thresholds larger than 0.005. Predicted probabilities below or equal to 0.5, indicated low gene expression rates and vice versa (Supplementary Data 7). EPM occurrence and predicted gene expression levels were linked using BLAMM[37]. EPMs were classified as conserved or mutated if present among all or absent among one genotype per gene, respectively (Supplementary Data 8). Boxplot depicting samples after bootstrap repetition using the 25th, 50th (median) and 75th percentiles along with the interquartile range, representing the central 50% of the data. Whiskers extend from the minimum to maximum values, showcasing the spread of the dataset. The two-sided F1 and Chi-squared test ($p$ value <0.0001) support statistical significance (Source Data). **d** Structural variations in the flanking regions altered gene expression measured in species indicated with asterisks for the exemplarily shown genes *Solyc02g087170* and *Solyc02g080300* (**g**). Congruently, high (red) and low (blue) probability scores of the multi-species reference (MSR) model indicate differential gene expression across the genotypes. Boxplot depicting sample characteristics using the 25th, 50th (median) and 75th percentiles along with the interquartile range, representing the central 50% of the data. Whiskers extend from the minimum to maximum values, showcasing the spread of the dataset. Outliers are depicted as dots (Source Data). **e** Gene maps of upstream regions around the TSS include UTR region (striped boxes), exons ("ATG" + filled boxes) and introns (white boxes), along with the location of significant EPM matches (e-value <0.00001) inside (white arrow) and outside (grey arrow) of their positional preference and the location of SVs (black arrows). **f** For closer inspection, only EPMs allocated to their preferred position were chosen. For sequence comparison, ITAG.3, representing genetic variant A and PAS104479, representing genetic variant B are displayed. EPMs of *S. lycopersicum* MSR model M006 p0m02 and M032 p0m15 lie within the 5'UTR of *Solyc02g087170*. An SV indel mutation of 178 bp located only 37 bp behind the TSS disrupts epmSoly-M006-p0m02 variant B genotype, starting 30 bp behind the expected TSS in genotypes of variant A. Due to the same SV, however, epmSoly-M032-p0m15 is now localised within its positional preferred range for species of genetic variant B. **g** Probability scores of the MSR model indicate differential gene expression for *Solyc02g080300* (See caption for sub-figure **b**). **h** (see caption for sub-figure **d**). **g** Within the first intron, 233 bp downstream of the expected TSS of *Solyc02g080300* epmSoly-M009-p0m04 was identified. An SV indel mutation of 10 bp lies 7 bp downstream of epmSoly-M009-p0m04, not disrupting the EPM. In contrast to the example before, the difference between two point mutations coincides with low probabilities of high gene expression for genotypes of variant A within epmSoly-M009-p0m04.

remarkable differences in phenotype and metabolism. The model, trained on a combined dataset from *A. thaliana*, *Z. mays* and *S. bicolor* achieved accuracy values of about 78% and 83% for *S. lycopersicum* and *S. pennellii*, respectively (shuffled-sequence control accuracy: *S. lycopersicum* = 51%, *S. pennellii* = 58%). The model performance was different for the negative and positive labels. For *S. lycopersicum* a true positive rate (TPR) of 0.9 and a true negative rate of 0.72 (TNR) indicated higher confidence for the positive predictions. This reflected the distribution of the expression predictions for genes of the "middle" second and third quartiles (Fig. 6a; yellow distributions). The same was observed for *S. pennellii*, with a TPR of 0.86 and a TNR of 0.81 (Fig. 6b).

To estimate if these MSR predictions are indicative of physiological and metabolic differences between the two species, we performed Mercator4 MapMan annotation of functional gene ontology on the *S. lycopersicum* and *S. pennellii* proteome[38]. Then, we expressed each functional category in terms of the observed (true labels) and predicted expression (prediction probabilities) of genes classified into each functional category. In total, 90 MapMan categories exhibited a significant change in the predicted expression between *S. lycopersicum* and *S. pennellii* (Wilcoxon rank-sum test; FDR <0.05; Fig. 6c and Supplementary Data 10). These included general terms such as 'enzyme' (3856 transcripts) and 'transcriptional regulation' (1829 transcripts), as well as specific ones, such as 'ZFP transcription factors' (26 transcripts) or 'protein translocation in chloroplast' (65 transcripts). The observed and predicted expression differences between *S. lycopersicum* and *S. pennellii* functional categories showed a moderate, but significant correlation (Spearman's ρ = 0.12; *p* value = 2.078e-11). This indicated that the MSR model successfully captures a differential expression of functional gene sets in the two tomato species. Almost all the significant categories were observed and predicted to be upregulated in *S. pennellii* with respect to *S. lycopersicum*. This concerns phytohormone action, external stimuli response, chromatin structure, lipid metabolism, as well as protein and RNA biosynthesis, and transcription regulation with their multiple sub-categories. Expression of several MapMan categories was overestimated by the MSR model, including e.g. chromatin structure, ubiquitin ligases and some TF families that indeed are very unlikely to be found in the upper quartile of expressed genes. Conversely, no MapMan categories were falsely predicted to be downregulated. The only category significantly downregulated in *S. pennellii* was the general term 'not annotated/not assigned'.

Finally, we zoomed in on enzymes of polyamine metabolism, as suggested by previous observations of strongly differential polyamine accumulation in *S. pennellii* and *S. lycopersicum* leaves[39] (Fig. 6d). Out of 40 genes annotated in *S. lycopersicum* 19 were observed as highly expressed (arginine decarboxylase ADC *Solyc01g110440, Solyc10g054440*; ornithine decarboxylase ODC *Solyc04g082030*, agmatine deiminase AIH *Solyc12g038970*; *N*-carbamoylputrescine amidase CPA *Solyc11g068540*; *S*-adenosyl methionine decarboxylase SAMDC *Solyc01g010050*, Solyc02g089610, *Solyc05g010420*; spermidine synthase SPDS *Solyc04g026030*; spermidine synthase SPMS *Solyc03g007240*; diamine *N*-acelytyltransferase SSAT *Solyc07g006340, Solyc08g006760, Solyc08g006765, Solyc08g068770, Solyc10g084640* and peroxisomal polyamine oxidase PAO2/3/4 *Solyc02g081390, Solyc07g043590, Solyc12g006370* covering each of the 12 reactions with at least one highly expressed isozyme transcript except the arginase ARG, polyamine oxidase PAO1 and PAO5. Ten of these were predicted correctly to be highly expressed (AIH *Solyc12g038970*; SAMDC *Solyc01g010050*; CPA *Solyc11g068540*; SPMS *Solyc03g007240;* SSAT *Solyc07g006340, Solyc08g006760, Solyc08g006765, Solyc10g008640*; PAO2/3/4 *Solyc02g081390, Solyc07g043590*) and no false positives were recorded, resulting in a prediction precision of 1 for the high expression. At the same time, out of three labelled as low expressed, all three were predicted to be low expressed (SSAT *Solyc07g015960, Solyc09g082260* and *Solyc12g096840*) with nine false negatives (precision = 0.6). Summarising, all pathway enzymes in *S. lycopersicum* were predicted to be highly expressed as at least one isozyme, except for the ARG, PAO1 and PAO5, which have not been observed as highly expressed in the corresponding transcriptome data either. Analogous results were obtained for *S. pennellii*, with the precision of high expression prediction of 1 (each positive expression prediction associated with respectively high or medium expression level; ARG2

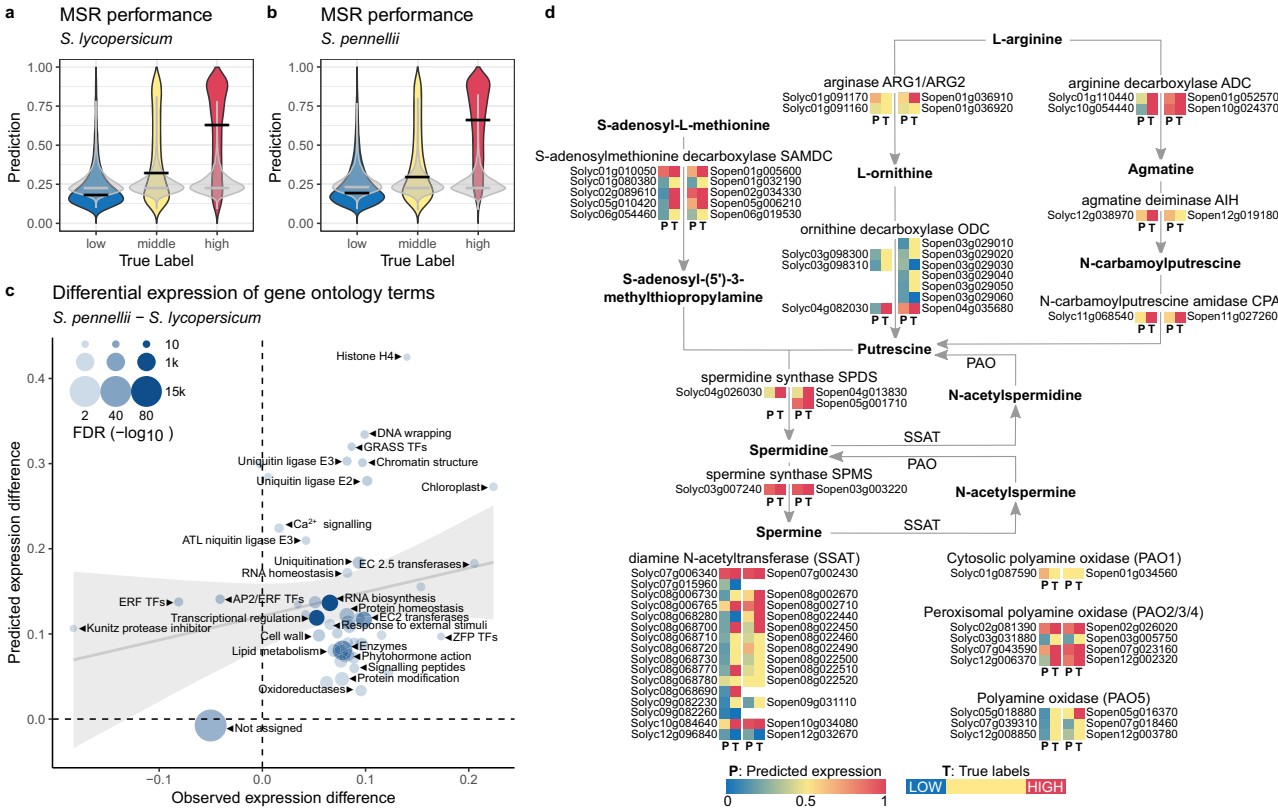

**Fig. 6 | Prediction and comparison of gene expression between *S. lycopersicum* and *S. pennellii* by the MSR model. a** Distribution of MSR model-predicted expression probabilities for the low (1st TPM quartile), middle (2nd and 3rd TPM quartiles) and high (4th TPM quartile) expressed transcripts of *S. lycopersicum*. Results of the shuffled-sequence control are plotted in grey for each respective expression level. **b** Distribution of MSR model-predicted expression probabilities for the low (1st TPM quartile), middle (2nd and 3rd TPM quartiles) and high (4th TPM quartile) expressed transcripts of *S. pennellii*. Results of the shuffled-sequence control are plotted in grey for each respective expression level. **c** Expression of MapMan functional categories exhibiting significant differential expression

between *S. lycopersicum* and *S. pennellii* (Wilcoxon rank-sum test FDR <0.05). The difference between mean expression for the observed and predicted values provides the x and y axis coordinates, respectively. Data were scaled so that −1 indicates expression in *S. lycopersicum* only, 0 indicates equal expression in both species and 1 signifies unique expression in *S. pennellii*. Linear regression represents fit with a PCC of 0.56 (PCC p value 5.827e-09, $R^2$ 0.32). **d** Mapping of predicted expression probabilities (P; predicted) and observed gene expression (T; true labels) for enzymes of the polyamine biosynthesis pathway in *S. lycopersicum* and *S. pennellii*. Heatmaps are sorted according to gene orthologs, with the *S. lycopersicum* on the left side and the respective *S. pennellii* orthologs on the right.

*Sopen01g036910*; ADC *Sopen01g052570, Sopen10g024370*; ODC *Sopen04g035680*; SAMDC *Sopen01g005600, Sopen02g034330, Sopen05g006210*; CPA *Sopen11g027260*; SPDS *Sopen05g001710*; SPMS Sopen03g003220; SSAT Sopen07g002430, Sopen08g002670, Sopen08g002710, Sopen08g022440, Sopen10g034080; PAO2/3/4 Sopen02g026020, Sopen07g023160, Sopen12g002320) and precision for low expression of 0.5 (true negatives obtained for ODC Sopen03g029030, Sopen03g029060; SSAT Sopen12g032670). Remarkably, despite the high sequence homology to *S. lycopersicum*, the *S. pennellii* orthologs showed distinct patterns of gene expression. This is especially visible for ARG, AIH SPDS, PAO5 and the SSAT enzymes. Additionally, ODC has been both observed and predicted to be highly expressed in *S. pennellii*, while AIH showed lower expression. In summary, the MSR model correctly indicated the polyamine biosynthetic pathway as highly expressed in both species, identified with high precision the isozymes that are most likely highly expressed for each reaction, and finally highlighted some differences between *S. lycopersicum* and *S. pennellii* in terms of expression of enzymes and their homologues.

## Discussion

In this study, we presented a new simple approach to the challenge of plant cistrome annotation. In contrast to other approaches e.g. screening for a very broad range of protein-DNA interactions[12],

identifying specific TF-DNA binding events[40,41], or extracting motifs enriched in genes exhibiting a stimulus response[42], here we focused on a specific aspect of cistrome function. Namely, we determined the level of gene expression, and built a mathematical model for robust and accurate prediction of it from DNA sequence.

There are notable features differentiating our approach from those used in most other studies on cistrome annotation. First, our study represents a genome-wide identification of predictive features related to the mRNA turnover and not those associated with the regulation of gene expression in response to any stimulus or tissue specificity. Second, while we highlighted some key DNA-TF binding events, a wider range of functional sequence features at the level of DNA and likely mRNA too (e.g. polyadenylation signals) was captured. In fact, most of the identified features were near the TSS and TTS sites, including gene UTRs (Fig. 3a and Supplementary Fig. 2). These regions are well known to be associated with mRNA turnover[43,44] and were recently targeted by synthetic biology approaches proving their central role in gene regulation, e.g. plant biotechnology chimeric 3' flanking regions strongly enhance gene expression in plants[45,46]. Further analyses will be required to characterise the molecular function of specific EPMs, e.g. the distinction between DNA−protein interaction (regulation) and RNA turnover or processing. Finally, due to the chosen training strategy, we avoided capturing sequence features conserved in all gene models (e.g. translation start and termination codons

or classic signals for transcription start[47]) but recovered only those predictive for high or low gene expression. While this provides a wealth of valuable and novel information, it is difficult to define a ground truth reference for our results; e.g. many structural and regulatory elements of the gene model are well-characterised and relatively easy to annotate[48,49], this is not yet the case for the turnover-related regulatory features[4].

Indeed, the majority of the identified sequence features, including three of seventeen complete EPM clusters and multiple 'in-cluster' EPMs, were not matching with previously described CREs despite their clear sequence and positional specificity. This has likely three main reasons. First, the incompleteness of the reference databases; the cognate TF-binding motifs might be uncharacterised or not included. Second, due to e.g. model regularisation, highlighted predictive sequences often represent only partial CREs. For example, important but redundant sequence features, e.g. palindromic and near-palindromic repeats, might be only partially captured. Finally, some EPMs were not TF-binding motifs but likely e.g. intron-, transposon- and UTR-associated sequence features, which are left out on prominent CRE databases.

The achieved model performance is rather striking, considering the complexity of plant gene regulation[50] and the relatively small training set, especially for the SSR models. In comparison to more complex models like the Enformer[17], which capture long-range interactions by scanning sequences longer than 100 kb, our models capture only the information present in the proximal cis-regulatory sequences. This makes them blind to the gene regulatory events being associated with very distal CREs, including e.g. regulation of tissue- development- and stress-specific genes[51]. Indeed, in our analysis of leaf and root samples, the models did not capture the tissue-specific regulation. The tissue-specific regulation seems to be independent of the genome-wide regulatory signature, highlighting also that prediction of low (or high) expression is more robust against confounding variables (Fig. 2), and illustrating the interaction between core cis-gene regulatory code and features rendering conditional gene expression.

This indicates that the core cis-regulatory landscape influencing the gene expression level in plants is relatively simple and very conserved. This is largely in agreement with the conservation of TF-binding sites in Arabidopsis thaliana ecotypes[20]. While the stimulus-induced changes in gene expression rarely translate directly to changes in the accumulation of respective proteins, the mRNA turnover, and the related amount of mRNA in the cytosol remains a key factor determining the general amount of respective protein product[52–54]. Thus, the determinants of the mRNA level are important elements of the system homoeostasis and are unlikely to exhibit high genetic variation[55,56]. The extent of that conservation is reflected by a comparison of the MSR model with the cross-species SSR performance. Increasing the training set for multiple species enabled the MSR models to capture that conserved regulation component and generalise very well to new genomes. It is also important to note that, at the same time, the MSR models remain relatively insensitive to species-specific features of gene models, such as varying lengths of regulatory sequences. In comparison, the random forest classifier trained on quantitative features of gene regulatory elements showed a good SSR performance, but not in the MSR scenario (Supplementary Fig. 4 and Supplementary Data 2).

One advantage of the parallel sequence feature and function annotation is the identification of the positional specificity of the EPM occurrence and function. In general, CREs may overlap with genic features like the UTRs, exons or introns, or are located distantly to the up- and downstream of the TSS or TTS, respectively[51]. In Arabidopsis thaliana, for example, several transcription factors are known to bind specifically at around 100 bp upstream of the TSS around the promoter[57]. Other studies show that certain CREs may preferentially occur in the first intron of A. thaliana genes[58]. While the ratio of CREs to

genes correlates, the distances in between likely correlate with the overall genome sizes due to mobility transposons or accumulation of repeats[59]. Previous experiments on reporter genes and comparative genomics in A. thaliana and Z. mays demonstrated changes in gene regulation for transcription factors binding sites situated within introns, where the distance between cognate CRE and the TSS was changed[60].

We demonstrated two EPMs that exhibit consistent positional preference and retained regulatory function, while also displaying variable positional preference that correlates with their regulatory activity across diverse flowering plant model organisms. In all four analysed species, A. thaliana, S. bicolor, S. lycopersicum and Z. mays, 2CWY+type EPMs exhibit strict positional preference of the transcription start site (TSS and TSS-downstream) and the transcription termination site (TTS and upstream-TTS). The 2CWY+ cluster is strictly associated with the prediction of low gene expression, too. EPMs of the 2GCB+ cluster, however, identified in A. thaliana, S. bicolor and Z. mays do occur rather up- or downstream of the TSS and TTS region, respectively or, in contrast, within the transcribed region, with conversely regulatory effects (Fig. 4d). These findings indicate that the linear distance of CREs to their cognate gene plays a defined role in gene regulation. Interestingly, it has also been demonstrated that most similar to EPMs 2CT+, CT(GA)-binding motifs of paralogous transcription factor BPC6 are localised at the TSS in barley (Hordeum) genes, as well[61]. These TFs are responsive to cytokinin stimulus[61], congruent with findings about BPC6 in response to brassinosteroids in A. thaliana[25]. Somehow contradictory to our findings, studies of paralogous transcription factor BPC1 in A. thaliana showed that by binding its cognate motif (GA-rich region) situated in a 5′-terminal intron is up-regulating gene expression[62]. The here identified 2CT+ type epmArth-S0-p1m05 appears localised in the 5′ transcript region but is associated with low rates of expression (Fig. 4e). In general, however, the EPMs identified here might not cover all regulatory elements. EPMs associated with more specific types of expression, e.g. developmental stages or under stress conditions, together with already characterised CREs in other organisms, will be identified in the future.

The predictive performance of EPMs from leaf SSR and MSR of A. thaliana was determined by measuring EPM occurrence within their preferred ranges. For genes with low expression levels, it is much less clear how these genes are correctly predicted from the occurrence of associated EPMs (Supplementary Data 6). This indicates that the CNN model may use further criteria for predictions like sequence context like combination of motifs. In contrast, however, EPMs associated with high rates of gene expression are strong predictors for high rates of gene expression, occurring in over 80% of matches in the correct expression class. Amongst them are the EPMs of the 2CT+ and RGA-GAR clusters that were found in root and leaf SSR and MSR models, respectively, underlining their regulatory relevance and evolutionary conservation.

Our analyses on the occurrence of EPMs and associated prediction of gene expression throughout different Solanum genotypes further shows how EPMs could be used for extended annotation and characterisation of genes (Fig. 5b–g). Our findings indicate that if indel or point mutations are present within the identified EPM or if the EPM is located outside of its typical positional preference, the CNN model disregards them for prediction. In addition, mutations in the EPM region do, in general, affect the model's prediction. Future characterisation of EPMs should consider both their sequence and positional preference and their association with genomic features such as UTRs, introns, transposons or repeats. The insertion of SVs within the cis-regulatory sequences has been associated with perturbations in gene expression, as shown for Solanum cultivars[36]. Genes that were predicted for differential gene expression and were found to undergo SV insertion together with log-fold changes in gene expression also all

had mutated EPM regions, as well (Fig. 5c). It will be highly interesting studying insertions of, e.g. transposable elements (TEs) in non-model plant Brassicaceae species, which were associated with perturbations in gene expression relevant to the transition from C3 to C4 photosynthesis[63]. Here, mining and characterisation of EPMs could improve the identification of the regulatory determinants in such evolutionary processes.

We have also demonstrated that our approach is able to highlight some genotype-determined phenotypic and metabolic features of newly sequenced plants. The MSR model highlighted known differences between *S. lycopersicum* and *S. pennellii* in terms of the increased biotic and abiotic stress resistance[64–66], growth[67,68] and cell wall[69]. This is striking considering that the compared species are homologous to each other, in comparison to the *A. thaliana*, *S. bicolor* and *Z. mays* genomes that were used to train the MSR model. The result suggests that the observed differential expression of gene functional groups results from changes in a relatively limited pool of conserved regulatory sequences. Application of the MSR model to the metabolic pathway of polyamine biosynthesis, i.e. compounds with highly differential accumulation in *S. lycopersicum* and *S. pennellii*[39], showed that the approach identifies active pathways (e.g. those with multiple reactions represented by at least one highly expressed enzyme), selects the most likely expressed isozymes for a given reaction, and highlights differences between the expression of orthologous genes in *S. lycopersicum* and *S. pennellii*. While these differences do not have to be directly associated with the observed differential accumulation of polyamines, the high precision and accuracy of expression prediction indicate that the MSR model is an efficient tool for increasing the confidence in gene candidate selection for molecular validation[70]. Namely, enzymes that are observed and predicted to be highly expressed based on a wealth of data across several species, might be a safer choice for follow-up characterisation than those that were observed to be expressed in a single experiment.

In summary, we showed that training a relatively simple deep learning model on publicly available large-scale genomic data provides a wealth of new biological information if combined with a proper model interpretation approach and set in an evolutionary context. This highlights the great potential of deep learning for high throughput cistrome annotation in newly sequenced genomes, as exemplified by our MSR models. Another important application is the prediction of the effects of genetic variation[71]. While current genomics approaches rarely enable functional annotation of the effect of polymorphisms[72,73] and rely mostly on statistical association, an increasing availability of long-read data enables making mechanistic links between observed variation and the downstream molecular network[36]. The presented example of gene expression across 14 tomato accessions represents an important step in this direction and opens perspectives for routine prediction pipelines for new genomes, tissues, and environmental scenarios. The ability to predict which genes are likely to be expressed and which are not might be also crucial for the selection of gene candidates in GWAS studies[74–76], or in the selection of functional enzyme homologues in reconstructed metabolic pathways[70]. While transcriptomic and metabolomic data are often used for that purpose[75,77–79], a purely sequence-based approach will significantly accelerate the process.

## Methods

### Reference genomes, proteomes, cDNA, gene models and data processing

The reference genomes, proteomes, cDNAs and gene models were downloaded from the Ensembl plants database (v52). Transcriptome data were downloaded from the National Center for Biotechnology Information (NCBI) Sequence Read Archive (SRA) database using the fasterq-dump (version 2.11.3). These were trimmed with sickle (version 1.33), mapped unto reference cDNA using Kallisto (version 0.46.2) and

Kallisto outputs were processed using the tximport package of the R statistical language to obtain the normalised counts per gene in transcripts per million (TPM) (Supplementary Data 1 and Source Data). To train the CNN models, the expression of transcript isoforms was aggregated to produce a single TPM value for gene expression.

Up- and downstream flanking sequences were extracted by anchoring at the gene start and end coordinates. Taking DNA strandedness into consideration, sequences from the negative strand were reverse complemented. The length of the input sequence data was estimated empirically in two rounds of model performance test. First, promoter and terminator sequences of different lengths were tested (500, 1000, 1500, 2000,2500, 3000 nt) while keeping the 5′ and 3′ UTRs of constant length of 500 nt. Secondly, we fixed the promoter and terminator lengths to 1000 bp and simultaneously varied the lengths of the UTRs (100, 300, 500, 700 nt). Based on the accuracy and auROC statistics (Supplementary Fig. 2 and Source Data) and computational complexity, 1000 nt promoter – 500 nt 5′UTR – 500 nt 3′UTR – 1000 nt terminator sequences combined a good performance and interpretability with reasonable model training and interpretation time; further extension of them did not result in significant improvements in any of the tested species. The range also captures more than half of the complete annotated UTRs across all species with the median UTR length between 135 nt for the *A. thaliana* 5′UTR to 378 nt for the *S. bicolor* 3′UTR (Supplementary Fig. 2). Accordingly, gene flanking regions were one-hot-encoded and for each gene combined into a single continuous sequence and separated by a 20 nt zero-padding between 5′ and 3′ UTR.

### Producing non-homologous training and validation sets

We built training and validation sets for the CNNs as binary classifiers to predict genes as either lowly or highly expressed. We used the thresholds of the lower and upper quartiles of the distribution of the log-transformed transcript per million values (<25% percentile, between 25% and 75% percentile, > 75% percentile, respectively). CNNs were trained using chromosomal level cross-validation, in which one chromosome was left out for validation and the rest used for training. To mitigate imbalance during chromosome-wise cross-validation during training, we randomly down-sampled the sequences of the majority class without replacement. Consequently, accuracy is utilised as the performance measure throughout the manuscript. The respective F1 scores are provided in Supplementary Data 2, 3 for comparison, applicable for unbalanced sampling.

To prevent overestimation of model performance through overfitting to evolutionary relatedness, we excluded genes on the validation chromosome that had homologues on the training chromosomes in SSR models. To produce non-homologous training and validation sets, we constructed local protein databases using proteomes for every species. The proteomes were blasted against their respective databases (Protein–Protein BLAST 2.9.0+). Two sequences were considered homologous if the blast hit had an e-value <0.001 and a bit score >50. Genes in the validation chromosome that had homologous pairs in the training chromosomes were dropped.

### Architecture of convolutional neural networks and training strategy

Convolutional neural networks were built using the sequential API of tensorflow-keras. All models had three convolutional blocks, each convolutional block had two convolutional layers followed by a max pooling and a dropout layer.

The final convolutional block was followed by a fully connected block with two fully connected layers interspaced by dropout layers. All dropout layers used a dropout rate of 0.25, all convolutional and fully connected layers used Rectified Linear unit activation functions. The output layer contained a single unit and the sigmoid activation function. Models were trained with a maximum number of 100 epochs.

To further mitigate overfitting, we used the *EarlyStopping callback* (patience = 10), the *ModelCheckpoint callback* was used to restore the best-seen model with the lowest validation loss. Finally, the *ReduceLROnPlateau callback* (patience = 5) was used to decrease the learning rate to improve model training.

Stochastic gradient descent, as implemented by the *Adam optimiser* (learning rate = 0.001), was used to take gentle steps towards the lowest achievable binary cross entropy loss.

## Model interpretation

We computed hypothetical and actual importance scores for all validation sets across all chromosomes. These scores are used in two ways: firstly, importance scores were averaged across all nucleotides per sequence to obtain single scores per nucleotide and then averaged across all sequences to obtain a single representation of the most salient regions. Secondly, the hypothetical and actual importance scores are used as inputs to TF-MoDISco, which extracts the predictive regions from the importance scores as seqlets. These seqlets are then clustered into groups and each group is aggregated into a representative motif. Due to memory and runtime constraints, we used the default value for the maximum number of seqlets per metacluster.

## EPM characterisation and comparison

We retrieved expression-predictive motifs (EPMs) using TF-MoDISco that returns both cluster-weighted models and position weight matrices (PWMs). PWMs were characterised with R statistical language and the package JASPAR2022 calculates various properties of PWMs, such as information content and consensus sequence[24]. In addition, we searched for potential hits between EPMs and already characterised TF binding motifs on the JASPAR plants database (Supplementary Data 4). With the algorithm implemented in the R package motifStack EPMs could be converted and clustered by homology using the Smith–Waterman algorithm[80] (Supplementary Data 4). EPMs extracted from the SSR model and MSR model with TF-MoDISco were mapped to their respective positional preferred range on the *A. thaliana* gene flanking region with BLAMM utilising a sensitivity e-value of 0.0001[37]. In avoidance of overlapping positional ranges lower and upper quartiles from the seqlet extraction and clustering step were used as margins to reduce outliers. In addition, more than one-tenth of all seqlets per motif must be found within the up and/or downstream region, else the region was excluded from mapping for the respective EPM. The occurrence of each EPM from metacluster 1 or 0 (association to low and high gene expression) was counted among the measured expression class of the gene (low, high) and the predicted expression class of the gene, using the respective model in R. To evaluate the overall predictive performance of each EPM, we calculated the motif's enrichment, using a variant of a formula proposed by Smet and colleagues[35]. This formula calculates the log2 fold change in the odds of an EPM associated with one gene expression class being present in genes with the same expression class compared to genes of the opposite expression class. Here, a positive enrichment results in a positive value and vice versa. In addition, we calculated percentages of EPM occurrences within respective expression classes and tested for significance from chi-square tests against normal distribution for the assumption that EPMs would be distributed to both equally (Supplementary Data 6). Scripts and code written for the extraction, characterisation, and comparison of EPMs were performed in RStudio 4.2.2 and BLAMM are provided in the repository https://github.com/NAMLab/DeepCRE[37,81].

## Investigation of flanking gene sequences of *S. lycopersicum* sub-variants

Candidate orthologous gene selection followed the identification of predicted perturbations in gene expression of fourteen wild and cultivated Solanum *spp*[36]. We selected 14 diverse genotypes for prediction: *S. pimpinellifolium* (PAS014479, BGV006775), *S. lycopersicum var. cerasiforme* (BGV006865, BGV007931, BGV007989), *S. lycopersicum processing* (M82), *S. lycopersicum fresh* (EA00371, Fla.8924, LYC1410) and *S. lycopersicum vintage* (PI69588, Brandywine, EA00990, PI303721). Perturbations in gene expression of wild and cultivated *Solanum spp.* were predicted using the multi-species model (MSR) and expressed as probabilities ranging from 0 to 1, predicting low and high gene expression (Supplementary Data 7).

We tested the occurrence of EPM variation states, conserved or mutated across the 15 *Solanum* genotypes, coinciding with predicted shifts between low and high levels of gene expression. The criteria for defining gene expression heterogeneity relied on the predicted MSR leaf models probability, with variance thresholds set above 0.005. Predicted probabilities below or equal to 0.5, indicated low gene expression rates and vice versa (Supplementary Data 7). To link EPMs with gene expression patterns, mapping was performed between EPMs from the *S. lycopersicum* MSR model and genes with predicted homogenous and differential gene expression rates across the fifteen different genotypes using BLAMM (e-value <0.0001)[37]. For each gene EPMs were classified as conserved or mutated if these were present across all genotypes or differed from *S. lycopersicum* ITAG 4.1, respectively (Supplementary Data 8). Addressing the imbalance in the number of genes with homogenous (n = 27,993) and predicted differential gene expression ($n$ = 2053), percentages conserved and mutated genes were calculated by randomly pooling genes to a set size of 100 and bootstrapping 1000 times. Scripts and code written in R 4.2.2 provided in the repository https://github.com/NAMLab/DeepCRE[81].

We found 385 genes that were predicted for differential gene expression, with structural variants (SVs) in their upstream or downstream 5 kbp regions, which were also detected for log-fold change in gene expression levels, according to Alonge and colleagues and containing mutated EPMs (Fisher exact test statistic value is <0.00001). Six genes from the fifteen genotypes were randomly selected for in-detail analyses aligned with MAFFT[82] (Supplementary Data 8, 9). EPMs extracted from the leaf MSR model of *S. lycopersicum* were mapped to the candidate gene alignments with BLAMM[37] (Supplementary Data 8). Up- and downstream gene regions were investigated manually for the presence of structural variants and other mutations within regions matching EPMs in their positional preferred ranges and annotated (Supplementary Data 9). The two examples, *Solyc02g08170.4* and *Solyc02g080300.3* (Fig. 5), were selected to explain the effects of EPM mutations on gene expression prediction, as both genes contain only one EPM region within its preferred range that was mutated, respectively.

## Functional annotation and pathway mapping

Functional annotation for comparison of the model accuracies between functional categories was performed de novo using Mercator4[38] on respective proteomes (Supplementary Data 11). Functional annotation of *S. lycopersicum* and *S. pennellii* genes to MapMan functional categories was also performed using Mercator4[38] on ITAG 4.1 and the latest *S. pennellii* genome Schmidt and colleagues (2017)[83], as described in Mercator4[38]. The significance of the shift of median between *S. lycopersicum* and *S. pennellii* shown in Fig. 6c was performed by a two-sided Wilcoxon rank-sum test with Benjamini–Hochberg $p$ value correction (Supplementary Data 10). Metabolic pathway mapping for polyamine biosynthetic pathway was done using MapMan 'bin 8' and the super-pathway of polyamine biosynthesis POLYAMINSYN3-PWY from the current TomatoCyc annotation v6.0.0. All statistics and visualisation were done using R 4.2.2 and Bioconductor[84].

## Reporting summary

Further information on research design is available in the Nature Portfolio Reporting Summary linked to this article.

## Data availability

The reference genomes sequence and annotations (*A. thaliana*, *S. lycopersicum*, *S. bicolor* and *Z. mays*) used for extraction of gene flanking regions and estimation of transcript profiles were downloaded from Ensembl plants database v52 (plants.ensembl.org) GCA_000001735.1 [https://plants.ensembl.org/Arabidopsis_thaliana], GCA_000188115.3 [https://plants.ensembl.org/Solanum_lycopersicum], GCA_000003195.3 [https://plants.ensembl.org/Sorghum_bicolor], and GCA_902167145.1 [https://plants.ensembl.org/Zea_mays]. Transcriptomic short-read data was downloaded from the National Center for Biotechnology Information (NCBI) Sequence Read Archive (SRA) database for leaf and root data from Bioprojects to determine transcript profiles PRJEB32665, SRP010775, PRJNA171684, PRJEB22168, PRJNA237342, PRJNA640858, PRJNA217523, and PRJNA271595. For the analyses of the fifteen *Solanum* genotypes, we used as reference sequences and annotations the Sol Genomics Network [https://solgenomics.net/ftp/genomes/]. For *Solanum pennellii* we used the accessions from Schmidt and colleagues (2017)[83] [http://www.plabipd.de/portal/solanum-pennellii]. These reference datasets were processed as described in the methods section to generate results. Source data are provided with this paper.

## Code availability

Custom code used in this study is available from the GitHub repository and Zenodo (https://github.com/NAMlab/DeepCRE)[81].

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

## Acknowledgements
The authors want to thank CEPLAS for the financing of S.M.Z., the IPK Gatersleben IT team for providing and maintenance of the computing cluster, and all members of the Network Analysis and Modelling lab at the IPK Gatersleben for discussions and suggestions. Dr. Simon Maria Zumkeller works at the Cluster of Excellence on Plant Sciences (CEPLAS), Heinrich-Heine-Universität Düsseldorf, 40225 Düsseldorf, Germany was funded by the Deutsche Forschungsgemeinschaft (DFG, German Research Foundation) under Germany´s Excellence Strategy—EXC-2048/1—project ID 390686111.

## Author contributions
F.F.P. performed bioinformatic analyses, designed and trained the CNN models, evaluated their performance, and performed feature selection. S.M.Z. contributed to the characterisation of EPMs and motif search in *Solanum* cultivars, A.S. and M.G. consulted the results and corrected the manuscript. J.S. designed and supervised the study and performed the functional enrichment and pathway analysis. F.F.P., S.M.Z., and J.S. designed figures and wrote the manuscript. All authors edited and approved the final manuscript version. F.F.P. and S.M.Z. contributed to the manuscript equally.

## Funding

## Competing interests
The authors declare no competing interests.
