## [Peer Review File · Nature Communications]

Deep learning the cis-regulatory code for gene expression in selected model plantsReviewer #1 (Remarks to the Author):

Gene expression prediction is the core issue for transcriptional regulation, especially in plants breeding. Because the prediction model combined with interpretability methods can help identify critical cis-regulatory elements (CREs) which significantly contributes target gene expression.

The authors developed deep learning models for gene expression prediction (classification models for predicting low or high expressions) on four model plant species. They achieved around 80% prediction accuracy in single-species reference (SSR) models and got 65-78% prediction accuracy in multi-species reference (MSR) models. Importantly, the authors employed a powerful interpretability tool of 'DeepLIFT' to assign base contribution score to each base and then used a TF motif analysis tool of 'TF-MoDISco' to identify Expression-Predictive Motifs (EPM). They found that EPMs showed a general specificity towards the 5'UTR and 3'UTR, which make sense in biological knowledges.

For model applications, the authors used a trained model (the MSR model trained with *Solanum lycopersicum* as the validation set) to predict differences in expression of genetic variants of two tomato genes of Solyc02g08170.4 and Solyc02g080300.3 on 15 wild and cultivated *Solanum* spp. Prediction results and EPM identifications were found matched with the genomic structural variants (SVs).

Overall, I think that the strategy using deep learning modeling and interpretability is the mainstream method for CRE identifications nowadays. However, the deep learning architecture the author used is too simple and lacks novelty, which hinders better prediction accuracy of gene expression. It is good for authors to use 'DeepLIFT' and 'TF-MoDISco' to perform EPM analysis, and I think this is the first contribution of the current work. Another contribution is the application of the trained model for identifying causal SVs that contributes high gene expression across 15 wild and cultivated *Solanum* spp. The manuscript is well organized. I have some queries as follows:

1. The choice of 1000nt upstream and 500nt downstream of the transcription start site (TSS), and 500nt upstream and 1000nt downstream of the transcription termination site (TTS) seems too subjective. The author should either cite relevant publications to support the choice, or provide comprehensive analysis about the reference genomes to support this choice.
2. In my opinion, the above lengths of promoter, 5'UTR, 3'UTR and terminator are not necessarily optimal. I strongly suggest that the authors perform a length optimization and get the optimal length with the data-driven strategy. Another important issue is that the optimal length is not necessarily the same for all four plant species, because the genome size is quite different. The authors should comprehensively discuss it.
3. For large genome, such as maize, there are many distal CREs that are dozens of Kb away from TSS. However, the current model cannot cover distal CREs, making the prediction accuracy limited.
4. In the middle of the manuscript, the authors introduced a random forest model with manual genomic features. Although the RF model has comparable prediction accuracy with the deep learning model, I think this section should be removed because it has not brought novel insights and it has no obvious connections with other sections in the whole manuscript.
5. The author should specify the output of the deep learning model. In the abstract, the author wrote 'link gene sequence data with mRNA copy number for the plant species', however after I carefully checked the details in 'Material and Methods', I found that the output is just the maximal TPM of RNA-seq analysis. Therefore, 'mRNA copy number' is not appropriate, and I suggest that the authors rename it.
6. Data availability. The authors should provide all the data and codes during training, testing and model applications for other researchers repeating their results.

Reviewer #2 (Remarks to the Author):

This manuscript aims to investigate the relationship between the DNA sequences in the proximal regions of the transcription start sites and transcription termination sites of any give gene and its

expression level across four plant species: *Arabidopsis thaliana*, *Sorghum bicolor*, *Solanum lycopersicum*, and *Zea mays*. The authors employed relatively simple convolutional neural network (CNN) models, achieving an overall accuracy of approximately 80%. These models utilized either species-specific references (SSR) or multi-species references (MSR) to predict gene expression levels for the same species or for a species that was excluded from the training data, respectively.

The findings of this study highlight the significance of the untranslated regions (UTRs) in determining gene expression levels. By employing various model interpretation techniques in the context of evolutionary relatedness, the authors identified conserved expression predictive motifs (EPMs) across the different species. The practical application of the models was demonstrated by utilizing the MSR models to accurately reveal expression differences in genes enriched in metabolic pathways that are known to vary between two related varieties.

While this research sheds light on the relationship between DNA sequences and gene expression levels in multiple plant species, I have several major concerns regarding the claimed findings and significance, detailed as follows:

1. One of the major findings of this study is the significance of UTR regions in determining mRNA abundance, which aligns with the previous research conducted by Mashburn et al. (2019) using multiple tissues from different developmental stages in maize. While the current study expands upon this finding by incorporating four plant species, it would be valuable to explore the importance and limitations of DNA sequence in the context of tissue-specificity. One possible approach to achieve this would be to conduct additional analyses that investigate the model's performance separately for tissue-specific genes versus non-specific genes. Furthermore, comparing the importance of genetic features in determining the expression levels would provide valuable insights into the role of DNA sequence in tissue-specific gene regulation.

2. The authors employed DeepLIFT saliency scores and TF-MoDISco to identify expression-predictive motifs (EPMs), which is an interesting approach. However, the current presentation of the observations and corresponding figures (L227-229, Figure 5a) lacks context and clarity. There is a need for a deeper exploration of these findings to provide a more comprehensive understanding. Furthermore, it would be beneficial to provide more details regarding the application of EPMs in predicting gene expression. Specifically, starting from L270 and Figure 5, it is unclear how the predictions were performed, what controls were used, and what statistical analyses were conducted to support the claim of robust performance. Additional information and analyses would strengthen the interpretation of the results and enhance the overall rigor of the study.

3. One application of the CNN models and EPMs was demonstrated by predicting and identifying differential gene expression among 14 sub-species and varieties of tomato, as described starting from L284 and shown in Figure 6. However, it is important to note that the manuscript primarily focuses on presenting a few exemplary genes, which restricts the overall outcome and generalizability of the findings. To strengthen the impact of CNN models and EPMs, it would be beneficial to present a more extensive set of genes in the results.

4. The clarity of the manuscript could be improved by addressing several aspects, such as the presentation of figures, figure legends, and the overall flow of the manuscript. I will provide a few examples, although these may not cover all possible cases:

- 4.1 It is unclear from the manuscript whether Figure 1b is a conceptual visualization or based on real data. To clarify this, it is important to provide additional information and make it clear to the readers. If Figure 1b is based on real data, the authors should specify the source of the data and provide the corresponding plots for the four species mentioned as Supplementary Figures.

- 4.2 Figure 1c may not be clear to general biologists without prior knowledge of CNN models. It would be helpful to provide additional explanation or context within the figure or in the main text to aid their understanding.

- 4.3 The legends for Figures 1-4 should be revised to provide sufficient information for readers to interpret and draw conclusions from the figures.

These are just a few specific examples, and I recommend reviewing the entire manuscript to identify areas where the presentation can be improved for better clarity and comprehension.

4.4 The section "Expression predictive motifs (EPMs) exhibit sequence and positional conservation" would be better placed immediately after the section "Identification and characterization of predictive sequence features." The current placement of the Random Forest section interrupts the flow of the discussion on EPMS.

Some minor points are listed as follows:

1. L109-114 : The CNN model used in this work is based on the pseudogene model with some modifications, as mentioned in the manuscript. However, it is noted that the specific details regarding these modifications and the rationale behind them are not explicitly described in the Materials and Methods section. It would be beneficial to include any limitations or challenges observed with the original pseudogene model and how the modifications were designed to address those issues. This will provide a clearer understanding of the specific changes made to the pseudogene model and how these changes improve its performance or adapt it to the current study's objectives.
2. L124-126: To prevent overfitting, the authors have taken measures to avoid including homologs in the training and validation sets. However, it is not explicitly mentioned in the manuscript how genes from the same family were handled in terms of overfitting. Considering that the authors are aware of the potential effect of gene families on overfitting, it would be valuable to provide additional information on how this issue was addressed.
3. Figure 3a: In agreement with the findings of Mashburn et al. (2019), the current study also observed that the saliency scores, or contribution scores, are highest within the gene body, specifically between the transcription start site (TSS) and transcription termination site (TTS). A discussion of this observation would be valuable in the manuscript. It could involve exploring the potential reasons and implications behind the observation. Does this suggest that the DNA sequences within this region play a crucial role in determining mRNA abundance?
4. L195-204: The authors determined 10 generic features to train Random Forest classifiers; however, the rationale behind the selection of these specific features was not provided in the manuscript. Providing a clear explanation of why these specific features were chosen will not only enhance the transparency of the study but also allow readers to better understand the underlying biological reasoning and potential implications of the selected features.
5. L383-407 (Figure 7d): To improve clarity in this paragraph, it would be beneficial to provide specific gene IDs within the context. For instance, in the sentence "Remarkably, six of these were predicted correctly to be highly expressed and no false positives were recorded, resulting in a prediction precision of 1 for the high expression," I recommend specifying which six genes were referred to. If space is a constraint, the information could be provided through indices or labels on the Figure panel. This will enhance the reader's understanding and provide a more comprehensive analysis of the results.

Responses to Reviewers

Thank you very much for the feedback on our manuscript. We look forward to having improved our work with your helpful remarks on the results and their presentation. We have added further text and supplements as detailed below. We include a correction-tracked version of our manuscript.

Reviewer 1

We are grateful for all the suggestions and comments from Reviewer 1. Addressing these remarks has, we believe, rendered our manuscript more accessible to researchers interested in the cis-regulatory landscape of plant genes, provided a stronger justification for the chosen model inputs, and enhanced the documentation of methods, thereby improving the reproducibility of the study.

1. The choice of 1000nt upstream and 500nt downstream of the transcription start site (TSS), and 500nt upstream and 1000nt downstream of the transcription termination site (TTS) seems too subjective. The author should either cite relevant publications to support the choice, or provide comprehensive analysis about the reference genomes to support this choice.

Response: We agree, we didn't explain this issue in the manuscript enough. We added additional explanations in the results section [II. 97-100 and II. 142-149] and Materials and Methods [II. 706-709]. The choice of the sequence range was based both on prior literature and our own testing of the model performance for different lengths. Similarly, to the previous study of Washburn et al (2019) this range was sufficient for the successful association of gene flanking regions to gene expression in *Z. mays*. As in Washburn et al (2019) we saw no significant improvement in model accuracy for promoters and terminators extended up to 3000 nt (**Supplementary Figure 1**). Concerning the literature support, the empirical estimation of the effective promoter length in Arabidopsis (Korkuc et al 2014) based on SNP frequency across 349 accessions of the 1.101 genomes collection hinted at average length of 500nt upstream of the TSS (**Revision Figure 1**) [taken from Korcuć et al. 2014]. Also, the location of elements of the core promoter, such as e.g., TATA-, GC-, CCAAT-boxes (reviewed e.g., in Brooks et al 2023) provides hints that this 500nt upstream region can be generalised across plant species, despite large differences in the genome and intergenic regions size. These findings convinced us that the chosen range is a safe estimate encompassing the conserved gene regulatory regions and being directly comparable with previous studies. We included respective references in the revised manuscript [II. 97-100]. The length of the UTR regions was based on length distribution of annotated UTRs (see response to comment #2).

Washburn JD, Mejia-Guerra MK, Ramstein G, Kremling KA, Valluru R, Buckler ES, Wang H. Evolutionarily informed deep learning methods for predicting relative transcript abundance from DNA sequence. *Proc Natl Acad Sci U S A*. 2019 Mar 19;116(12):5542-5549. doi: 10.1073/pnas.1814551116. Epub 2019 Mar 6. PMID: 30842277; PMCID: PMC6431157.

Paula Korkuć, Jos H.M. Schippers, Dirk Walther, Characterization and Identification of cis-Regulatory Elements in Arabidopsis Based on Single-Nucleotide Polymorphism Information, *Plant Physiology*, Volume 164, Issue 1, January 2014, Pages 181–200, <https://doi.org/10.1104/pp.113.229716>

Emily G. Brooks Estefania Elorriaga Yang Liu James R. Dudit Guoliang Yuan Chung-Jui Tsai Gerald A. Tuskan Thomas G. Ranney Xiaohan Yang Wusheng Liu. Plant Promoters and Terminators for High-Precision Bioengineering. *BioDesign Res*. 5;2023:0013. DOI:10.34133/bdr.0013

Revision Figure 1. (Figure 2 of Korkuć et al. 2014) - Average SNP density profile relative to the TSS. SNP densities (window size of 25 nt) were computed for all intergenic regions (black solid line) for case 1 (Fig. 1), SNPs only (collinear orientation of neighbouring genes, where the genes have the same orientation; dark gray solid line), and case 3, SNPs only, in which genes are oriented head-to-head (light gray solid line). A logistic curve with $f(x) = A + (B - A) / (1 + C \exp(2 Dx))$, where $A = 0.019$, $B = 0.015$, $C = 0.223$, and $D = 2 \cdot 0.0098$, was fitted to the all-SNPs data set (black dashed line).

2. In my opinion, the above lengths of promoter, 5'UTR, 3'UTR and terminator are not necessarily optimal. I strongly suggest that the authors perform a length optimization and get the optimal length with the data-driven strategy.

Another important issue is that the optimal length is not necessarily the same for all four plant species, because the genome size is quite different. The authors should comprehensively discuss it.

This is a very good suggestion. In fact, we performed an initial length optimization but didn't report it in full in the initial submission. In the revised version we extended that test and reported in the main text and Supplemental Figures. We divided the comment and our response into two parts:

“I strongly suggest that the authors perform a length optimization and get the optimal length with the data-driven strategy. “

Response: In the revised version of the manuscript, we provided results of the comprehensive testing of different genomic ranges taken as inputs for the analysis (Materials & Methods section [II. 704-719] (**Supplementary Figure 1**)). We tested the performance of the model on all four species extending the sizes of the promoter and terminator while keeping the UTR regions of constant 500 nt length. We observed no significant improvement of the performance with increasing size of the input sequence. Secondly, we performed a similar test on the UTR regions while keeping the promoter and terminator sequences at a constant 1000 nt length. Similarly, results displayed no significant improvement of the performance.

“(…) the optimal length is not necessarily the same for all four plant species, because the genome size is quite different. The authors should comprehensively discuss it.”

Response: This is indeed a very important issue. To address it we provided estimates of model performance for different lengths of the input sequences (**Supplementary Figure 1**) and added a respective paragraph to the Materials and Methods section [II. 704-719]. We also compared the distribution of the length of the annotated UTRs between the four species and highlighted the percentage of the annotated UTRs fully covered by the selected nucleotide range [II. 172-175] (**Supplementary Figure 2**). While the percentage of the covered UTR's varied remarkably, increasing the input UTR range beyond 500bp did not translate to improvement of model performance. Thus, we concluded that most predictive features locate within the selected range for all species of interest. At the same time, we showed that the length of the UTR regions is indeed predictive towards gene expression. Our results on training the random forest (RF) classifier on ten quantitative features of regulatory sequences indicated high importance of 3'UTR length (moved in the revised version of the manuscript to the **Supplementary Materials, Supplementary Data 1 and Supplementary Figure 4**). Additionally, the discrepancy of performance between the RF-SSR models and RF-MSR models strongly indicate the importance of species-specific importance (**Supplementary**

Figure 4). We added discussion of these results also in the main text [II. 578-583].

3. For large genomes, such as maize, there are many distal CREs that are dozens of Kb away from TSS. However, the current model cannot cover distal CREs, making the prediction accuracy limited.

Response: We agree that identification of regulation related to expression induced by phytohormone-interacting CREs or modulated by enhancers would require much larger ranges, which would likely differ between the species of interest. This would also require different model architecture and training strategy (e.g., Enformer model; Avsec et al. 2021). In fact, as a response to a related request of reviewer 2, we trained additional models on gene expression in root tissues. We observed that models were not able to capture tissue-specificity of the expression of $\cong 5\%$ genes differentially classified in roots and shoots of the respective species, while performed equally well for the conserved ones (**Supplementary Figure 5**). This indicated that the regulation is driven by additional factors, e.g., those located outside of the taken flanking region. Regulatory effects resulting from long range interaction (*in trans*) could not be captured. At the same time however, taking the proximal flanking regions enable achieving high and comparable performance across different tissues RNA-seq profiles.

Our analysis indicated that the selected region: a) provides the same performance as larger flanking ranges (**Supplemental Figure 2**) while keeping the models simpler and thus more interpretable, b) includes the 500 nt minimum known to contain majority of plant promoters elements [II. 100-102], and c) assures that intergenic regions are covered, which is especially relevant in *Arabidopsis thaliana* with an average intergenic sequence length of 1579 (see table below). These points are supported by distribution of the importance scores, as well (**Figure 3**). The importance scores 500 nt outside of the gene boundaries are marginal.

In summary, we expect a variance and “imperfect” model performance that cannot fully reflect gene regulation. This is mainly due to *in trans* regulatory interactions, with e.g., distal enhancing or silencing CREs and others may include e.g., variation in trans-gene regulatory network, epigenetic effects and noise. Including more potential regions, where such elements may be located, does however, whether improve performance and complicates interpretability. We added a comment on this issue in the discussion section [II. 555-566].

file	Total ncDNA sequence (bp)	Total gene sequence (bp)	Total IGS sequence (bp)	Average_igs_size (bp)
Arabidopsis_thaliana.TAIR10.56.gff3	119,146,348	67,736,680	51,409,668	1579.503

Sorghum_bicolor.Sorghum_bicolor_NCBIv3.55.chr.gff3	683,645,045	126,676,128	556,968,917	15,774.58
Solanum_lycopersicum.SL3.0.55.chr.gff3	807,224,664	148,740,922	658,483,742	18,792.88
Zea_mays.Zm-B73-REFERENCE-NAM-5.0.55.chr.gff3	2131846805	182707673	1949139132	44850.07

4. In the middle of the manuscript, the authors introduced a random forest model with manual genomic features. Although the RF model has comparable prediction accuracy with the deep learning model, I think this section should be removed because it has not brought novel insights and it has no obvious connections with other sections in the whole manuscript.

Response: The section has been moved to and rewritten in Supplemental Materials addressing comment #2.

5. The author should specify the output of the deep learning model. In the abstract, the author wrote ‘link gene sequence data with mRNA copy number for the plant species, however after I carefully checked the details in ‘Material and Methods’, I found that the output is just the maximal TPM of RNA-seq analysis. Therefore, ‘mRNA copy number’ is not appropriate, and I suggest that the authors rename it.

Response: We replaced “mRNA copy number” with “quantitative gene expression” both in the abstract [I. 23] and in the discussion section [II. 521-523].

6. Data availability. The authors should provide all the data and codes during training, testing and model applications for other researchers repeating their results.

Response: We improved annotation of the code (GitHub <https://github.com/NAMlab/DeepCRE>) and provided fully processed read count input data (Supplementary table 1).

Reviewer 2

We appreciate all the suggestions from Reviewer 2. By addressing these remarks, the scope of our study has been broadened to include both leaf and root tissues, allowing us to delve into and discuss tissue-specific gene regulation. This expansion

together with additional statistics has also enhanced the utility of the identified regulatory motifs and, we hope, improved the clarity of the manuscript.

1. One of the major findings of this study is the significance of UTR regions in determining mRNA abundance, which aligns with the previous research conducted by Washburn et al. (2019) using multiple tissues from different developmental stages in maize. While the current study expands upon this finding by incorporating four plant species, it would be valuable to explore the importance and limitations of DNA sequence in the context of tissue-specificity. One possible approach to achieve this would be to conduct additional analyses that investigate the model's performance separately for tissue-specific genes versus non-specific genes. Furthermore, comparing the importance of genetic features in determining the expression levels would provide valuable insights into the role of DNA sequence in tissue-specific gene regulation.

Response: To address this issue, we performed a new analysis of root gene expression in all four species for the revised manuscript. We trained and tested the root model in the same way as we did for the leaf samples and compared the model performance separately for tissue-specific genes (defined as genes classified differentially in leaf and root samples according to the strategy used in the study; **(Figure 2); (Supplementary Table 2) [ll. 155-161 and 177-184]**). For the 427 (*A. thaliana*), 572 (*S. lycopersicum*), 693 (*S. bicolor*) and 623 (*Z. mays*) DEGs, the result indicates that our model does not capture regulation related to tissue-specificity, providing the accuracies close to 0.5 for the tissue-specific genes for all four species **(Supplementary Figure 4)**. This is largely determined by the fact that organ-specificity was not the objective of the model training. Considering the low contribution of organ-specific genes to the total pool of labelled genes their contribution is minor (between 4% genes in *A. thaliana* and 6% in *Z. mays*). At the same time, the low prediction accuracy for root- and leaf- specific genes indicated that tissue-specific gene regulation is indeed a confounding variable and is not captured by the model. A dedicated strategy, including good quality standardised tissue-specific expression data, probably larger gene flanking regions, training based on fold-changes and multiclass classification model would be likely required to address the tissue-specificity.

2. The authors employed DeepLIFT saliency scores and TF-MoDISco to identify expression-predictive motifs (EPMs), which is an interesting approach. However, the current presentation of the observations and corresponding figures (L227-229, Figure 5a) lacks context and clarity. There is a need for a deeper exploration of these findings to provide a more comprehensive understanding. Furthermore, it would be beneficial to provide more details regarding the application of EPMs in predicting gene expression. Specifically, starting from L270 and Figure 5, it is unclear how the predictions were

performed, what controls were used, and what statistical analyses were conducted to support the claim of robust performance. Additional information and analyses would strengthen the interpretation of the results and enhance the overall rigor of the study.

Response: To address this point we have edited text in the results and discussion section, **Figure 4, Supplementary Table 3** and added a new **Supplementary Table 4 and Supplementary material 1** to the manuscript. The reviewer's comment concerns context and clarity of our findings outlined in the section of **II. 226-245** and **Figure 4a**, namely, finding and using expression predictive motifs (EPMs). We addressed this point by a more detailed explanation and depiction of exemplarily chosen EPM. We describe how the importance scores are the deciding metric for the models and how EPMs are a generalisation (aggregation of seqlets) of these. Accordingly, we added the sum, minimum and maximum contribution score of each of the 522 characterized EPMs in **Supplementary Table 3**. In an updated version of **Figure 4** we have added panel (a) showing the estimated contribution scores for *A. thaliana* gene AT1G01650. Within this gene upstream region, we could find epmArth-S019-p0m09. Regions that match epmArth-S019-p0m09 have positive contribution scores. Hopefully, this example gives clearer context, how importance scores are calculated (also shown in **Figure 3**) and applied to EPMs. Consequently, EPMs are generalized used features by the models crucial to prediction and further analyses, e.g., cross-species comparison.

To underline relevance and interpretability of EPMs, we reduced the number of shown or described examples. We now focus on epmArth-S019-p0m09 of the CT+ type cluster over the older ones displayed in old **Figure 4**. We choose epmArth-S019-p0m09 because it has the highest sum importance score measured in this study, shows high enrichment within its associated expression class, underscoring good predictive performance and matches the well characterized TFBS of factor BP5. Introducing this case now early in the result section, in line with respective changes to **Figure 4** should provide more clarity and context [**II. 236-245, 262-265 and 321-328**]. In addition, we added **Supplementary data 2** to make our nomenclature proposal and clustering EPMs more accessible. This file contains an alignment of all consensus sequences of EPMs of the leaf SSR and MSR models.

To address the reviewer's comment regarding statistical analyses supporting the predictive performance of EPMs we have conducted a specified analysis. We introduced a new section to the results and on that purpose: **"The EPMs of metacluster 0 of *A. thaliana* SSR and MSR leaf model are predictors for high levels of gene expression "** [**II. 329-365**]. We also made complementary statements in discussion [**II. 621-630**]. We further changed the section "EPM characterization and comparison" [**line 755-780**] in the Material and Methods accordingly [**line 764-785**].

Before we described that EPMs found with the models occur in restricted ranges in close proximity to the gene boundaries (**Supplementary Table 3**). Accordingly, we then exemplarily mapped EPMs of *A. thaliana* SSR and MSR leaf model to their positional preferred ranges within *A. thaliana* genome, respectively and calculated their enrichment within the measured and predicted classes, as well (**Supplementary Table 4**). For this calculation we used a formula proposed by Smet and colleagues (2023) that equally measured the occurrence of a distinct sequence motif within a defined expression class (Smet et al. 2023). We found that, most notably, *A. thaliana* EPMs of the SSR and MSR leaf model associated with metacluster 0, can be used as predictors for high gene expression classes. This, however, is not the case for EPMs of metacluster 1, respectively. Here, enrichment scores indicate that these EPMs are equally distributed to both expression classes. Accordingly, we weakened our previous wording regarding the EPMs predictive performance “robustness”. Instead, we now directly refer to the enrichment score or sum importance score of an EPM when we discuss its interpretability, see new section [II. 236-245, 262-265 and 321-328].

Due to the changes made in **Figure 4** the references for the panels had been changed accordingly throughout the manuscript.

Smet, D., Opdebeeck, H. & Vandepoele, K. Predicting transcriptional responses to heat and drought stress from genomic features using a machine learning approach in rice. *Front. Plant Sci.* **14**, 1212073 (2023).

3. One application of the CNN models and EPMs was demonstrated by predicting and identifying differential gene expression among 14 sub-species and varieties of tomato, as described starting from L284 and shown in Figure 6. However, it is important to note that the manuscript primarily focuses on presenting a few exemplary genes, which restricts the overall outcome and generalizability of the findings. To strengthen the impact of CNN models and EPMs, it would be beneficial to present a more extensive set of genes in the results.

Response: To address the reviewers comment we clarified and generalized our strategy, how to identify gene variants with differential gene expression and associated EPMs. For this, we edited text in the results and discussion section, edited **Figure 5** and added new **Supplementary Table 5**, **Supplementary Table 6** and **Supplementary Data 3** to the manuscript.

To clarify how we found candidate genes from the fifteen tomato genotypes we added a small section in the results [II. 388-399]. Here we describe a strategy that filtered for the intersections of genes between the dataset of Alonge et al 2021 and Peleke et al 2023. Alonge and colleagues (2021) detected structural variation within 5 kb in flanking genes regions and linked these with log fold changes in gene-expression across the 15 different genotypes. In our manuscript we predicted gene expression classes. In total, we identified an intersection of genes with log fold

changes and SVs and predicted differential gene expression of 385 cases (**Figure 5b**). From this pool, we added five more examples to the new **Supplementary Data 3** besides the two examples already shown in now **Figure 5d-i**. To underline the model's sensitivity towards mutation among genotypes, we investigated the more general relation between the occurrence of EPMs and prediction of gene expression classes. We classified genes as homogeneously and differentially expressed and measured the occurrences of EPMs among these (**Figure 5c**). Regions that match EPMs among genes with homogenous gene expression are about three times more often conserved than in genes with differentially gene expression. Regions that match EPMs in genes with differential expression are about two times more often mutated. Due to the changes made in **Figure 5** the references for panels had been changed accordingly.

Please find a novel section added to “Investigation of flanking gene sequences of *Solanum lycopersicum* sub-variants” [**line 793-808**].

Alonge, M., Wang, X., Benoit, M., Soyk, S., Pereira, L., Zhang, L., Suresh, H., Ramakrishnan, S., Maumus, F., Ciren, D., Levy, Y., Harel, T. H., Shalev-Schlosser, G., Amsellem, Z., Razifard, H., Caicedo, A. L., Tieman, D. M., Klee, H., Kirsche, M., Aganezov, S., Rhyker Ranallo-Benavidez, T., Lemmon, Z. H., Kim, J., Robitaille, G., Kramer, M., Goodwin, S., Richard McCombie, W., Hutton, S., Van Eck, J., Gillis, J., Eshed, Y., Sedlazeck, F. J., van der Knaap, E., Schatz, M. C. & Lippman, Z. B. Major Impacts of Widespread Structural Variation on Gene Expression and Crop Improvement in Tomato. *Cell* **182**, 145–161.e23 Preprint at <https://doi.org/10.1016/j.cell.2020.05.021> (2020)

In this manuscript, we are eager to highlight the potential of deep learning models for delving into gene variants at the nucleotide level and identifying their impact. While we recognize the value of using individual EPMs as predictors for gene expression levels (**Supplementary Table 4**) and the sum contribution scores of EPMs as partial interpretable tools (**Supplementary Table 3**), they represent only a limited facet of the intricate saliency maps (exemplarily shown in **Figure 4a**) associated with individual genes. Although these criteria aid in understanding model predictions to some extent, deep learning models offer a more holistic analysis of the genes, likely using contextual features, too.

For instance, features such as UTR length, which are identified by our Random Forest (RF) analyses (**Supplementary material text 1 and Supplementary Figure 3**), could potentially be integrated into our Convolutional Neural Network (CNN) models but are not encompassed within the EPMs. Therefore, to comprehensively address the reviewers' point and achieve our long-term research goals, indeed, we recognize the need for a distinct experimental setup. The data we used from Alonge et al., 2021, though valuable, may not be sufficient for this approach, particularly due to differences between log fold changes in gene expression and measured or predicted changes in defined expression classes.

In conclusion, we appreciate the reviewers' valuable input and acknowledge that their suggestions have sparked a deeper exploration of our research. We are committed to pursuing a more comprehensive approach to uncover the intricate connections between genetic variants and gene expression by using deep learning and exploration of EPMs in the future.

4. The clarity of the manuscript could be improved by addressing several aspects, such as the presentation of figures, figure legends, and the overall flow of the manuscript. I will provide a few examples, although these may not cover all possible cases:

4.1 It is unclear from the manuscript whether Figure 1b is a conceptual visualisation or based on real data. To clarify this, it is important to provide additional information and make it clear to the readers. If Figure 1b is based on real data, the authors should specify the source of the data and provide the corresponding plots for the four species mentioned as Supplementary Figures.

Response: In general, we went thoroughly through the manuscript and tried to optimize clarity in text for the overall flow and presentation of figures. In particular, we addressed the aforementioned remark by adding more detailed descriptions of Figures 1, 2 and 3. See for Figure 1 [II. 833-853], for Figure 2 [II. 861-880] and Figure 3 [line 882-394]. The text for Figure 1b now reads: “Genes were assigned into low (dark orange), medium (blue) and high (red) expression classes based on the upper and lower quartile of the logMaxTPM distributions (orange, blue and red) exemplarily shown for *A. thaliana* here. Histograms for transcript profiles of *S. lycopersicum*, *S. bicolor*, and *Z. mays* are shown in **Supplementary figure 1**. The threshold values for leaf transcript profiles of *A. thaliana*, *S. lycopersicum*, *S. bicolor*, and *Z. mays* were 0.529, 0.000, 0.151, and 0.431 for the lower and 1.635, 1.217, 1.391, and 1.793 for the higher quartile, respectively (**Supplementary Table 1**).”

“

4.2 Figure 1c may not be clear to general biologists without prior knowledge of CNN models. It would be helpful to provide additional explanation or context within the figure or in the main text to aid their understanding.

Response: We added the explanation of the model architecture and the function of each of its elements to the results section [II. 109-122].

4.3 The legends for Figures 1-4 should be revised to provide sufficient information for readers to interpret and draw conclusions from the figures. These are just a few specific examples, and I recommend reviewing the entire

manuscript to identify areas where the presentation can be improved for better clarity and comprehension.

Response: We revised both the legends and the figures for improved clarity. See detailed comment on this for reviewers 1 point 4.

4.4 The section "Expression predictive motifs (EPMs) exhibit sequence and positional conservation" would be better placed immediately after the section "Identification and characterization of predictive sequence features." The current placement of the Random Forest section interrupts the flow of the discussion on EPMs.

Response: Thank you for pointing this out. We moved the Random Forest section (as suggested by Reviewer 1, as well) to the Supplementary Materials (Supplementary Data 1 and Supplementary figure 4).

1. L109-114 : The CNN model used in this work is based on the pseudogene model with some modifications, as mentioned in the manuscript. However, it is noted that the specific details regarding these modifications and the rationale behind them are not explicitly described in the Materials and Methods section. It would be beneficial to include any limitations or challenges observed with the original pseudogene model and how the modifications were designed to address those issues. This will provide a clearer understanding of the specific changes made to the pseudogene model and how these changes improve its performance or adapt it to the current study's objectives.

Response: We have added the extended explanation of the model architecture in the results section and commented on the changes in respect to the use used in Washburn et al. (2019) study [II. 117-122].). Notably, we didn't change the pseudogene model, but the model architecture. As indicated, changes made concerned using 1D convolutional layers instead of the 2D convolutional layers with a max pooling layer. The change is a major simplification of the model and thus reduction of the parameter space without loss in performance. Such model structure has been the choice of recent studies working on genomic sequences (e.g., Enformer, Avsec et al 2021). The model of Washburn and colleagues is more reminiscent of that classically used for the 3 channels of an RGB image. While this might be suitable for integration of multi-modal data, we don't have multiple channels in the input sequence data, and therefore decided for a simpler model.

Avsec, Ž., Agarwal, V., Visentin, D. et al. Effective gene expression prediction from sequence by integrating long-range interactions. Nat Methods 18, 1196–1203 (2021). <https://doi.org/10.1038/s41592-021-01252-x>

Avsec, Ž., Weilert, M., Shrikumar, A. et al. Base-resolution models of transcription-factor binding reveal soft motif syntax. *Nat Genet* 53, 354–366 (2021). <https://doi.org/10.1038/s41588-021-00782-6>

2. L124-126: *To prevent overfitting, the authors have taken measures to avoid including homologs in the training and validation sets. However, it is not explicitly mentioned in the manuscript how genes from the same family were handled in terms of overfitting. Considering that the authors are aware of the potential effect of gene families on overfitting, it would be valuable to provide additional information on how this issue was addressed.*

Response: We have reworded and expanded the mentioned fragment for clarity. We prevent overestimation of the model performance due to evolutionary relatedness by omitting genes in the validation set that have homologs in the training set stated in the results section [II. 130-134] and Materials and Methods [II. 720-726].

3. Figure 3a: *In agreement with the findings of Washburn et al. (2019), the current study also observed that the saliency scores, or contribution scores, are highest within the gene body, specifically between the transcription start site (TSS) and transcription termination site (TTS). A discussion of this observation would be valuable in the manuscript. It could involve exploring the potential reasons and implications behind the observation. Does this suggest that the DNA sequences within this region play a crucial role in determining mRNA abundance?*

Response: We agree that we didn't highlight this finding enough. We added respective comments in the discussion section [II. 530-532].

4. L195-204: *The authors determined 10 generic features to train Random Forest classifiers; however, the rationale behind the selection of these specific features was not provided in the manuscript. Providing a clear explanation of why these specific features were chosen will not only enhance the transparency of the study but also allow readers to better understand the underlying biological reasoning and potential implications of the selected features.*

Response: We added an extended explanation of the choice of the selected features and provided respective citations. To address the request of Reviewer 1 we moved the Random Forest classifier section to Supplementary Materials.

5. L383-407 (Figure 7d): *To improve clarity in this paragraph, it would be beneficial to provide specific gene IDs within the context. For instance, in the sentence "Remarkably, six of these were predicted correctly to be highly expressed and no false positives were recorded, resulting in a prediction*

precision of 1 for the high expression," I recommend specifying which six genes were referred to. If space is a constraint, the information could be provided through indices or labels on the Figure panel. This will enhance the reader's understanding and provide a more comprehensive analysis of the results.

Response: We added respective Solyc and Sopen IDs to the paragraph [II. 474-515].

Reviewer #1 (Remarks to the Author):

The authors have addressed all the problems that I raised.

Reviewer #2 (Remarks to the Author):

In the revised manuscript, the authors have thoughtfully addressed the concerns I previously raised, leading to an overall improvement in the quality of the manuscript. Their explanations regarding the data source and analysis pipelines have provided valuable insights into a couple of critical questions.

First, I observed that the genome sequence for Arabidopsis was obtained from the Ensembl Plant database (v52), presumably for Col-0. However, not all of the leaf transcriptome datasets used were generated using Col-0. For example, SRX1734407 includes samples from the ecotype Tiesha-9. It remains unclear why the authors opted to use the Col-0 genome sequence to predict the expression levels of Tiesha-9.

Second, the comprehensive source data has drawn my attention to the fact that, for each species, the transcriptome datasets are sourced from a single SRP study. This raises the importance of exploring model performance within species, utilizing datasets generated from leaves in various SRP projects. It is reasonable to expect that the model should demonstrate robustness against technical variations stemming from factors such as leaf age, different handlers, and growth conditions.

Thirdly, I was unable to locate the figure legends for all of the supplementary figures. The information provided in the manuscript is not sufficient for me to comprehensively assess the supplementary figures.

Additional minor issues are presented as follows:

m1. (L106-108)

The author's use of quantiles to classify gene expression is not explicitly clear regarding whether they considered the root and shoot together or separately. Based on the figures, my interpretation leans towards the combination of root and shoot data. However, I have concerns that this approach might lead to the exclusion of root- or shoot-specific genes.

m2. (L126-127)

The model performs optimally on genes with expression levels below the lower quantile or above the upper quantile. However, it's not specified if this corresponds to the 25% or 75% quantile. Given that low expression is a common filter in many analyses, it would be more relevant to the scientific community if a similar filter were also first applied. Additionally, it would be informative to explore the nature of these genes through enrichment analysis to determine if they are associated with specific functional terms.

Furthermore, it seems that Figure 2a may not align with the context being discussed. The statement regarding dividing the data into more classes and training a regression model resulting in considerably lower accuracies (Figure 2a) doesn't appear to be in sync with the preceding discussion.

m3. (L135-137)

As the model performance shows no significant difference with input sequences ranging from 500 to 3000 nt of the promoter and terminator, it suggests that the core regulatory elements are likely located within the 500 bp range. Therefore, it would be valuable for the authors to conduct downstream analyses using 500 bp as input sequences and subsequently compare and discuss the results in relation to those obtained using 3000 bp input sequences.

Additionally, I encountered difficulty in locating the legends for ALL the Supplementary Figures, which hindered my ability to interpret the statistics presented in Figure S2. Clarifying the legends for all Supplementary Figures would be greatly appreciated.

m4. (L437-445) The general applicability of the MSR models is examined using the cultivated *S. lycopersicum* and its wild-type relative *S. pennellii*. The accuracies were only calculated for MSR models, I think shuffled-sequence controls are needed as negative controls.

Reviewer #2 (Remarks to the Author):

In the revised manuscript, the authors have thoughtfully addressed the concerns I previously raised, leading to an overall improvement in the quality of the manuscript. Their explanations regarding the data source and analysis pipelines have provided valuable insights into a couple of critical questions.

Thank you very much for the constructive feedback on our manuscript. We are especially thankful for the major comment 1. By addressing it we avoided an inconsistency in the input data and slightly improved the performance of the models. We made corrections, and have added further text and supplements as detailed below.

Major comment 1: *First, I observed that the genome sequence for Arabidopsis was obtained from the Ensembl Plant database (v52), presumably for Col-0. However, not all of the leaf transcriptome datasets used were generated using Col-0. For example, SRX1734407 includes samples from the ecotype Tiesha-9. It remains unclear why the authors opted to use the Col-0 genome sequence to predict the expression levels of Tiesha-9.*

Response: Thank you very much for pointing out this critical issue. Indeed, some of the training RNA-seq data for leaf samples used accessions mismatched the reference genome used for mapping. This mistake has been corrected, and expectedly the model performance proved stable. We observed a slight improvement in model accuracy (e.g. *Arabidopsis thaliana* acc_{leaf} accuracy increased from 83.79% to 85.59%) likely reflecting the improved sequence-to-expression concordance. The increased performance was observed also for the MSR models. Respectively, feature selection resulted in very similar saliency score profiles and provided 260, instead of 261 EPMS, with 79,27% and 99,43% matching the previous ones with PCC-similarity larger than 95% and 90%. The changes have been introduced across the manuscript text and figures, most importantly: lines 158-163, 182-189, 224-226, 508-512, 525, Figures 2, 3, 4, 5, 6

Major comment 2: *Second, the comprehensive source data has drawn my attention to the fact that, for each species, the transcriptome datasets are sourced from a single SRP study. This raises the importance of exploring model performance within species, utilising datasets generated from leaves in various SRP projects. It is reasonable to expect that the model should demonstrate robustness against technical variations stemming from factors such as leaf age, different handlers, and growth conditions.*

Response: This is a valid point, while showing the performance of our models across different species, tissues and accessions, indeed we didn't provide the comparison between individual studies that might be related to batch effects. We have now addressed this concern by adding **Supplementary Figure 6**. In this figure, control experiments for leaves with identical genomes and transcriptome data were compared to different experiments where [1] new models were trained and [2] models were cross-validated/tested. For training, we used the classification of logMaxTPMs from PRJNA237342 and PRJNA237342, which comes from alternative leaf transcriptome preparations in *A. thaliana*, similar to the control experiments shown within this study.

Changes in the manuscript can be found at line 168:

"RNAseq profiles for the same species and condition can differ due to technical and experimental variations. Accordingly, we exemplarily trained and cross evaluated leaf SSR models from multiple comparable RNAseq experiments of *A. thaliana*. These achieved high test performances supporting our methods reproducibility (**Supplementary Figure 6**)".

Major comment 3: *Thirdly, I was unable to locate the figure legends for all of the supplementary figures. The information provided in the manuscript is not sufficient for me to comprehensively assess the supplementary figures.*

Response: We are not sure about the reason for this, as we provided all legends in the electronic submission system. To be sure all the captions are available, in the revised manuscript we provide the Supplementary Figures caption text together with each Supplementary Figure.

Additional minor issues are presented as follows:

Minor comment 1 (L106-108): *The author's use of quantiles to classify gene expression is not explicitly clear regarding whether they considered the root and shoot together or separately. Based on the figures, my interpretation leans towards the combination of root and shoot data. However, I have concerns that this approach might lead to the exclusion of root- or shoot-specific genes.*

Response: The RNAseq profile data of the separate tissues (root/leaf) has been passed into the training pipeline individually. This indeed might not have been clearly stated. We edited the sentence in **line 108**.

Minor comment 2 (L126-127): *The model performs optimally on genes with expression levels below the lower quantile or above the upper quantile. However, it's not specified if this corresponds to the 25% or 75% quantile. Given that low expression is a common filter in many analyses, it would be more relevant to the scientific community if a similar filter were also first applied. Additionally, it would be informative to explore the nature of these genes through enrichment analysis to determine if they are associated with specific functional terms.*

Furthermore, it seems that Figure 2a may not align with the context being discussed. The statement regarding dividing the data into more classes and training a regression model resulting in considerably lower accuracies (Figure 2a) doesn't appear to be in sync with the preceding discussion.

Response: Indeed, in the manuscript we refer to a lower and upper quartile, referring to the 25% lowest expressed genes and 25% of the highest expressed genes. For clarity, we added this information (**lines 108-109**) to the revised version of the manuscript.

We agree that most RNA-seq data applies a minimal gene expression filter before the data analysis, and this is what the community might be familiar with. In our study however we avoided such filtering because it requires setting a threshold that is dependent on the sensitivity of the analysis (size of the library and genome, quality of the genome annotation etc), being individual for every study. Therefore the distinction between low- and non-expressed genes would be difficult to address appropriately. Instead, in the revised version of the manuscript we provide a new **Supplementary figure 5** with the distributions

of predictions for each consecutive 5% percentiles, showing that the model performs similarly for low- and non-expressed genes. We hope this addresses the issue.

In order to address the relationship between gene function and prediction accuracy, in the revised version of the manuscript, we performed de-novo Mercator4 functional annotation for all genomes of interest (new **Supplementary Data 4**) and provided accuracy estimates for each individual functional category (new **Supplementary Table 3**). We added the respective results (**lines 881-883**) and methods (**lines 183-189**) to the revised version of the manuscript.

Minor comment 3 (L135-137): *As the model performance shows no significant difference with input sequences ranging from 500 to 3000 nt of the promoter and terminator, it suggests that the core regulatory elements are likely located within the 500 bp range. Therefore, it would be valuable for the authors to conduct downstream analyses using 500 bp as input sequences and subsequently compare and discuss the results in relation to those obtained using 3000 bp input sequences. Additionally, I encountered difficulty in locating the legends for ALL the Supplementary Figures, which hindered my ability to interpret the statistics presented in Figure S2. Clarifying the legends for all Supplementary Figures would be greatly appreciated.*

Response: In order to address this issue we performed DeepLIFT analysis for models with different promoter/terminator and UTR regions length and compared the resulting saliency maps (new **Supplementary Figure 8**). As for the model performance, the minimum of 500 nt range has been established for the promoter/terminator regions and UTRs, below which the performance has dropped. Consistently with the model performance, the downstream analysis of ranges up to 3000 bp indicated change of the saliency map only for ranges below 500 bp. Extension of the promoter/terminator range beyond that did not affect the saliency map. In the revised manuscript we address this issue in **lines 240-244** and **Supplementary Figure 8**.

Minor comment 4 (L437-445): *The general applicability of the MSR models is examined using the cultivated *S. lycopersicum* and its wild-type relative *S. pennellii*. The accuracies were only calculated for MSR models, I think shuffled-sequence controls are needed as negative controls.*

Response: In the revised version of the manuscript we provide violin plots for shuffled sequences in the modified **Figure 6**, and we added the accuracies for shuffled controls to the respective results section (**lines 491-492 and 497**).

Reviewer #2 (Remarks to the Author):

It is gratifying to observe the positive impact of my previous comments on enhancing the manuscript's quality. I appreciate the authors for incorporating datasets from consistent ecotypes and including additional datasets for testing. I acknowledge that introducing new comments at this stage may not be anticipated, and I regret not addressing this critical point earlier.

Throughout the manuscript, the authors have used accuracy as the metric to evaluate model performance. Accuracy assesses the correct classification of observations using the formula $(TP+TN)/(TP+TN+FP+FN)$. However, accuracy is not suitable for imbalanced problems, as achieving a high score is possible by simply classifying all observations as the majority class.

In this study, gene expression was categorized into low (less than 25% percentile), medium (between 25% and 75%), and high (above 75%), representing an imbalanced dataset. For imbalanced datasets, the F1 score, which combines precision and recall into a single metric by calculating the harmonic mean between them, would be a more appropriate measure.

I regret not highlighting this aspect in my previous feedback. I strongly believe that using the proper metric is crucial to substantiate the reported findings in this manuscript.

Reviewer #2 (Remarks to the Author):

It is gratifying to observe the positive impact of my previous comments on enhancing the manuscript's quality. I appreciate the authors for incorporating datasets from consistent ecotypes and including additional datasets for testing. I acknowledge that introducing new comments at this stage may not be anticipated, and I regret not addressing this critical point earlier.

Throughout the manuscript, the authors have used accuracy as the metric to evaluate model performance. Accuracy assesses the correct classification of observations using the formula $(TP+TN)/(TP+TN+FP+FN)$. However, accuracy is not suitable for imbalanced problems, as achieving a high score is possible by simply classifying all observations as the majority class.

In this study, gene expression was categorised into low (less than 25% percentile), medium (between 25% and 75%), and high (above 75%), representing an imbalanced dataset. For imbalanced datasets, the F1 score, which combines precision and recall into a single metric by calculating the harmonic mean between them, would be a more appropriate measure.

I regret not highlighting this aspect in my previous feedback. I strongly believe that using the proper metric is crucial to substantiate the reported findings in this manuscript.

Response to Reviewer #2

We acknowledge that the F1 measure is more suitable for imbalanced datasets. However, in our study, we actually balanced the training datasets by randomly down-sampling the majority class without replacement, hence our use of accuracy. In the attached revised version of the manuscript we added the F1 macro, F1 micro and F1 weighted scores computed using the scikit-learn to the **Supplementary Table 2** (as additional sheets). We also added respective F1 scores to the **Supplementary Table 3** for individual MapMan bins. Estimated F1 scores are congruent with the prior estimated model accuracy values.

We referenced the F1 scores in the results section (lines 163-164):

"(...) performance of all models and controls with F1 scores provided in Supplementary Table 2."

We modified the **Material and Methods** subsection *Producing nonhomologous training and validation sets* for clarity in terms of balancing the training datasets (lines 735-746):

"To mitigate imbalance during chromosome-wise cross-validation during training, we randomly downsampled the sequences of the majority class without replacement. Consequently, accuracy (Acc) is utilised as the performance measure throughout the manuscript. The respective F1 scores are provided in supplementary table 2 and 3 for comparison, applicable for unbalanced sampling."

In addition, we made minor changes in the following section to match continuity (see ll. 747)

Reviewer #2 (Remarks to the Author):

The authors have addressed the concerns that I raised.